# Model-free inference of direct network interactions from nonlinear collective dynamics

Jose Casadiego[1,2], Mor Nitzan[3,4,5], Sarah Hallerberg[2,6] & Marc Timme[1,2,7,8]

The topology of interactions in network dynamical systems fundamentally underlies their function. Accelerating technological progress creates massively available data about collective nonlinear dynamics in physical, biological, and technological systems. Detecting direct interaction patterns from those dynamics still constitutes a major open problem. In particular, current nonlinear dynamics approaches mostly require to know a priori a model of the (often high dimensional) system dynamics. Here we develop a model-independent framework for inferring direct interactions solely from recording the nonlinear collective dynamics generated. Introducing an explicit dependency matrix in combination with a block-orthogonal regression algorithm, the approach works reliably across many dynamical regimes, including transient dynamics toward steady states, periodic and non-periodic dynamics, and chaos. Together with its capabilities to reveal network (two point) as well as hypernetwork (e.g., three point) interactions, this framework may thus open up nonlinear dynamics options of inferring direct interaction patterns across systems where no model is known.

[1] Chair for Network Dynamics, Institute for Theoretical Physics and Center for Advancing Electronics Dresden (cfaed), Technical University of Dresden, 01062 Dresden, Germany. [2] Network Dynamics, Max Planck Institute for Dynamics and Self-Organization (MPIDS), 37077 Goettingen, Germany. [3] Racah Institute of Physics, The Hebrew University, 9190401 Jerusalem, Israel. [4] Department of Microbiology and Molecular Genetics, The Hebrew University, 9112001 Jerusalem, Israel. [5] School of Computer Science, The Hebrew University, 9190401 Jerusalem, Israel. [6] Fakultät Technik und Informatik, Hamburg University of Applied Sciences, 20099 Hamburg, Germany. [7] Bernstein Center for Computational Neuroscience (BCCN), 37077 Goettingen, Germany. [8] Department of Physics, Technical University of Darmstadt, 64289 Darmstadt, Germany. Correspondence and requests for materials should be addressed to J.C. (email: jose.casadiego@tu-dresden.de) or to M.T. (email: marc.timme@tu-dresden.de)

The collective dynamics of many natural systems ranging from regulatory circuits and metabolic systems[1–7] to communication, distribution, and supply networks[8,9] is derived from the direct interactions of their parts. Determining how such systems are connected may help us in understanding and controlling their function[10,11]. Current nonlinear dynamics approaches may recover direct interactions from the collective dynamics of a system if a mathematical model is provided in advance and only their unknown parameters, network links, and nonlinear terms are to be determined[11–19]. Such models, however, are usually not at hand under most experimental conditions, thereby constraining the applicability of these methods to a limited number of examples. Recent works[20,21] on low-dimensional systems suggest that approximating the dynamics through expansions in basis functions may reveal the interaction patterns, if such dynamics admits a sparse representation in the proposed basis. A more recent model-free approach that takes into account the nonlinear network dynamics requires to externally drive the systems in a controlled way, thus enabling reconstruction from experimental settings for one particular range of settings[22]. Common model-free approaches not considering the nonlinear system dynamics construct functional links by detecting statistical dependencies (e.g., correlations, mutual information, Granger causality, and extensions thereof)[23–31] and thus are prone to recover indirect interactions among the units of a network, for instance, due to common external inputs or decorrelating effects induced by other units in the network[11,27,28,32–34]. Although latest efforts have focused on filtering indirect connections[27,28] from pairwise statistical dependencies, recent studies show that these functional links can only match direct connections under specific homogeneity conditions[35], which rarely occur in real-world systems.

In this article, we propose a novel concept for inferring direct interactions in coupled dynamical systems, relying only on their nonlinear collective dynamics, with neither assuming specific dynamic models to be known in advance nor assuming the dynamics admits a sparse representation, nor imposing controlled drivings, nor expecting statistical dependencies to faithfully reveal direct, physical interactions. To achieve this goal, we here change the perspective and ask which units $j$ of the network provide direct physical interactions to a given unit $i$ and appear on the right hand side of its differential equation, rather than asking for details of the interaction functions among those units. We demonstrate that the problem of inferring direct interactions based on observed nonlinear dynamics may be posed as a multivariate regression problem by introducing an explicit dependency matrix and thereby systematically decomposing each units dynamics into pairwise, three-point, and higher-order interactions with other units in the network. Such decompositions provide restricting equations for mapping the collective dynamics to direct interactions. We validate and characterize the predictive power of our approach by successfully revealing the structure of generic as well as specific biological model systems. These model systems may exhibit complex noisy dynamics such as transient dynamics toward steady states, periodic and non-periodic dynamics, or chaos, and have standard pairwise as well as hypernetwork (such as three point) interactions. Interaction networks may even be revealed if some units are not measured (and thus hidden during observation).

## Results

**Mapping time series to direct interactions**. To understand which information a time series contains about the direct interactions in networks, consider a system whose time evolution is given by

$$\mathbf{x} = \mathbf{F}(\mathbf{x}(t)) + \boldsymbol{\xi}(t) \qquad (1)$$

where $\mathbf{x}(t) = [x_1(t), ..., x_N(t)]^\in \mathbb{R}^N$ is the state of the entire system consisting of units with variables $x_i(t)$, $\mathbf{x} = d\mathbf{x}(t)/dt$ denotes its temporal derivative, $\boldsymbol{\xi}(t) = [\xi_1(t), ..., \xi_N(t)] \in \mathbb{R}^N$ represents external noise acting on the whole system, and $\mathbf{F} : \mathbb{R}^N \to \mathbb{R}^N$ is any smooth, typically nonlinear function that we assume to be unknown. Common examples are the regulation functions in models for gene regulatory networks[3,5,7] or rate laws in metabolic systems[36].

Given a multivariate time series

$$x_{i,m} := x_i(t_m) \qquad (2)$$

recorded at discrete time points $t_m = m\Delta t + t_0$, system identification aims to reveal the exact functional form of $\mathbf{F}$ and to exactly predict the systems future[37]. Owing to the high dimensionality of most networks, such identification is typically restricted or even impossible. Here we address the problem in a slightly yet essentially different manner, asking only: which of the variables $x_j$ directly acts on a given unit $i$ and thus explicitly appears on the right hand side of Eq. (1)? We aim to reveal not only pairwise network interactions, specified by terms of the form $\dot{x}_i = ... + g_j^i(x_j) + g_{ij}^i(x_i, x_j) + ...$, but also higher-order hypernetwork interactions, induced, for instance, by terms of the form $\dot{x}_i = ... + g_{jk}^i(x_j, x_k) + g_{ijk}^i(x_i, x_j, x_k) + ...$ where two or more units $j$ and $k$ different than $i$ jointly influence unit $i$ directly.

To distinguish among units and at the same time treat all orders of interactions simultaneously, we introduce explicit dependency matrices $\Lambda^i \in \{0, 1\}^{N \times N}$, diagonal matrices defined by

$$\Lambda_{jj}^i = \begin{cases} 0 & \text{if } \frac{\partial F_i}{\partial x_j} \equiv 0 \\ 1 & \text{if } \frac{\partial F_i}{\partial x_j} \not\equiv 0 \end{cases}. \qquad (3)$$

Hence, if a unit $j$ directly acts on unit $i$, we have $\Lambda_{jj}^i$ equals 1, and $\Lambda_{jj}^i$ equals 0 otherwise. With this notation, the dynamics of the units becomes

$$\dot{x}_i = f_i(\Lambda^i \mathbf{x}(t)) + \xi_i(t), \qquad (4)$$

where $f_i : \mathbb{R}^N \to \mathbb{R}$ is a smooth function that specifies the deterministic evolution of component $i$ and $\xi_i(t) \in \mathbb{R}$ represents external noise acting on $i$.

The explicit dependency matrix $\Lambda^i$ selects which variables $x_j$ directly control the rate of change of $x_i$, thus going beyond the related graph-theoretical notions of adjacency and incidence matrices and thereby emphasizing aspects of the dynamics: first, it offers a uniform representation of pairwise and higher-order interactions; and second, it is thus suitable for generic dynamical systems representations, as it appears exactly once in the right hand side of Eq. (4).

The resulting generic model (Eq. (4)) links state space points $\mathbf{x}$ ($t$) at time $t$ to their rate of change $\dot{x}_i(t)$. In particular, the complemented system state $s_i(t) = [\mathbf{x}(t), \dot{x}_i(t)]^\in \mathbb{R}^{N+1}$ is an element of a higher-dimensional "dynamics space" $\mathcal{D}_i$ for each $i$ formed by the state space and the rate of change of unit $i$. Therefore, the $f_i$ specifying the dynamics defines a smooth manifold $\mathcal{M}_i \subset \mathcal{D}_i$, with the $\Lambda_{jj}^i$ indicating whether or not $\mathcal{M}_i$ is constant in direction $x_j$. cf. Fig. 1.

In practical scenarios, the functions $f_i$ are generally not accessible. We address this challenge in two stages. First, we functionally decompose the dynamics of units $i \in \{1, 2, ..., N\}$ into

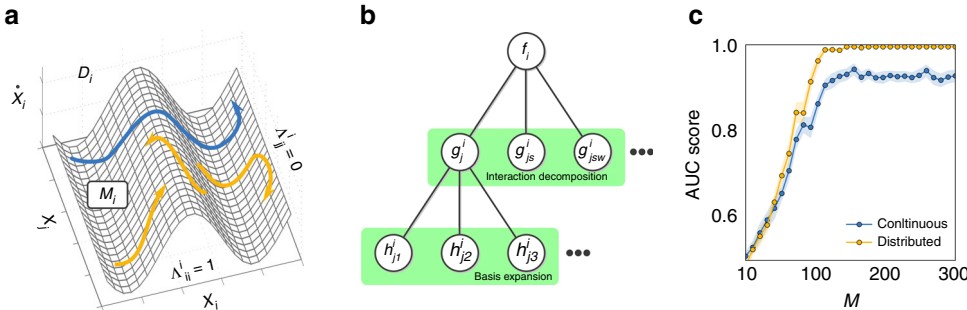

**Fig. 1** Sampling dynamics spaces to reveal network structure. **a** Eq. (4) determines a mapping from state space components **x** to rates of change $\dot{x}_i$, defining a smooth manifold $\mathcal{M}_i$ (in dynamics space $\mathcal{D}_i$) determined by $f_i$ and $\Lambda^i$. **b** Two-stage decomposition of unknown functions $f_i$, first into interactions of different orders with other units in the network, second each interaction into basis functions, thereby resulting in a linear system (Eq. (6)) restricting the connectivity structure. **c** Quality of inferences versus the number $M$ of measurements (displayed here for non-periodic dynamics of networks of phase-coupled oscillators of $N = 20$ and $n_i = 10$ incoming connections per unit). "Continuous sampling" takes as observed dynamics one long trajectory of $M$ sample time steps; "distributed sampling" takes $M/m$ short time series of $m = 10$ sample time steps starting from different initial conditions, creating a longer time series of length $M$. The initial conditions were randomly drawn from uniform distributions defined in the interval $[-\pi, \pi]$

interaction terms with the entire network as

$$\dot{x}_i = f_i(\Lambda^i \mathbf{x}) = \sum_{j=1}^{N} \Lambda_{jj}^i g_j^i(x_j) + \sum_{j=1}^{N} \sum_{s=1}^{N} \Lambda_{jj}^i \Lambda_{ss}^i g_{js}^i(x_j, x_s)$$
$$+ \sum_{j=1}^{N} \sum_{s=1}^{N} \sum_{w=1}^{N} \Lambda_{jj}^i \Lambda_{ss}^i \Lambda_{ww}^i g_{jsw}^i(x_j, x_s, x_w) + \ldots + \xi_i, \quad (5)$$

where $g_j^i : \mathbb{R} \to \mathbb{R}$, $g_{js}^i : \mathbb{R}^2 \to \mathbb{R}$, $g_{jsw}^i : \mathbb{R}^3 \to \mathbb{R}$ and, in general $g_{j_1 j_2 \ldots j_K}^i : \mathbb{R}^K \to \mathbb{R}$, represent the (unknown) $K$-th order interactions between units $j_k$ for all $k \in \{1, 2, \ldots, K\}$ and unit $i$. Specifically, the decomposition (Eq. (5)) separates contributions to unit $i$ arising from different orders, e.g., pairwise and higher-order interactions with other units in the system. The $\Lambda_i$ are defined such that, if $\Lambda_{rr}^i \equiv 0$, all functions $g_{j_1, j_2, \ldots, j_K}^i$ with any of the indices $j_k = r$ disappear from the right hand side of Eq. (5).

Given that functions $g_{j_1, j_2, \ldots, j_K}^i$ are taken to not be accessible, we decompose each $g_{j_1, j_2, \ldots, j_K}^i$ into basis functions $h$ as

$$\dot{x}_i = \sum_{j=1}^{N} \Lambda_{jj}^i \sum_{p=1}^{P_1} c_{j,p}^i h_{j,p}(x_j) + \sum_{j=1}^{N} \sum_{s=1}^{N} \Lambda_{jj}^i \Lambda_{ss}^i \sum_{p=1}^{P_2} c_{js,p}^i h_{js,p}(x_j, x_s)$$
$$+ \sum_{j=1}^{N} \sum_{s=1}^{N} \sum_{w=1}^{N} \Lambda_{jj}^i \Lambda_{ss}^i \Lambda_{ww}^i \sum_{p=1}^{P_3} c_{jsw,p}^i h_{jsw,p}(x_j, x_s, x_w) + \ldots + \xi_i, \quad (6)$$

where $P_k$ indicates the number of basis functions employed in the expansion, c.f. ref. [38]. Thus, provided a time series (2) where $\Delta t$ is sufficiently small such as to reliably estimate time derivatives $\dot{x}_{i,m}$, revealing direct interactions becomes identifying the non-zero coefficients in the right hand side of Eq. (6) that best fit the estimated $\dot{x}_{i,m}$. Such expansions (Eq. (6)) differ qualitatively from those developed in refs. [15,16,18,20] since ours do neither require the functions $g_{j_1 j_2, \ldots, j_K}^i$ to be represented exactly by the basis functions chosen nor the condition to admit a sparse representation in the basis. Instead, we only require the functions $h$ to form any basis of a relevant function space, thereby additionally allowing the investigator to choose basis functions not appearing explicitly in any of the $g^i$. In particular, this reduced requirement implies that, for instance, all coefficients $c_{j,p}^i \equiv 0$ are (indistinguishable from) zero for all $p$ if there is no functional dependency $g_j^i(x_j) = \sum_p c_{j,p}^i h_{j,p}(x_j) \equiv 0$.

This weaker requirement is sufficient to impose a structure of blocks of zero and non-zero coefficients in Eq. (6), representing absent and existing interactions, respectively, thereby posing a mathematical regression problem with grouped variables[38–42]. To solve such structured problems, we developed the Algorithm for Revealing Network Interactions (ARNI) (Supplementary Note 1), a greedy approach based on the Block Orthogonal Least Squares (BOLS) algorithm[40]. Specifically, our approach takes the time series of all units in the network as inputs and returns a ranked list of interactions indicating the order in which interactions in the right hand side of Eq. (6) were identified as most strongly lowering a cost function (see text below and Supplementary Note 1/Supplementary Fig. 3). We remark that here we do not intend to recover the actual functional form of interactions, but instead we aim at determining the existence or absence of interactions between units. So, even if our scheme infers an optimal model from a given time series, it is not guaranteed that such a model would agree with an actual model generating the dynamics[43]. Indeed, the fact that we only ask for the units interacting with a given unit and not for details of the coupling functions enables robust performance across systems (compare Figs. 1, 2, 3, 4, 5, and 6).

**Revealing direct links in model systems**. To demonstrate the robustness of our approach, we inferred the interactions of model systems and compared our results to those obtained from thresholding correlations[11,44], partial correlations[45], and transfer entropy[46]. In particular, we have selected such quantities because they are model independent, and they have been traditionally used to quantify interactions in networked systems. We tested our framework on systems displaying diverse types of collective dynamics, such as transient dynamics toward steady states, non-periodic dynamics, and chaotic and noisy dynamics, as emerging in models of Michaelis Menten kinetics in gene regulation, generic heteroclinic, and generic chaotic oscillatory dynamics. We measured the quality of reconstruction in terms of area under the receiver-operating-characteristic curve (AUC) score (Supplementary Note 3). The AUC score equals 1 for perfect reconstruction and it equals 1/2 for predictions equivalent to random guessing.

Predictions improve with longer time series as well as by composing one long time series out of different short ones, as illustrated for non-periodic dynamics in Fig. 1c. This indicates that sampling sufficient parts of state space is essential for revealing direct network interactions. Generally, we found that if long time series are not available (or not preferred, see the following), compositions of short time series are at least equally

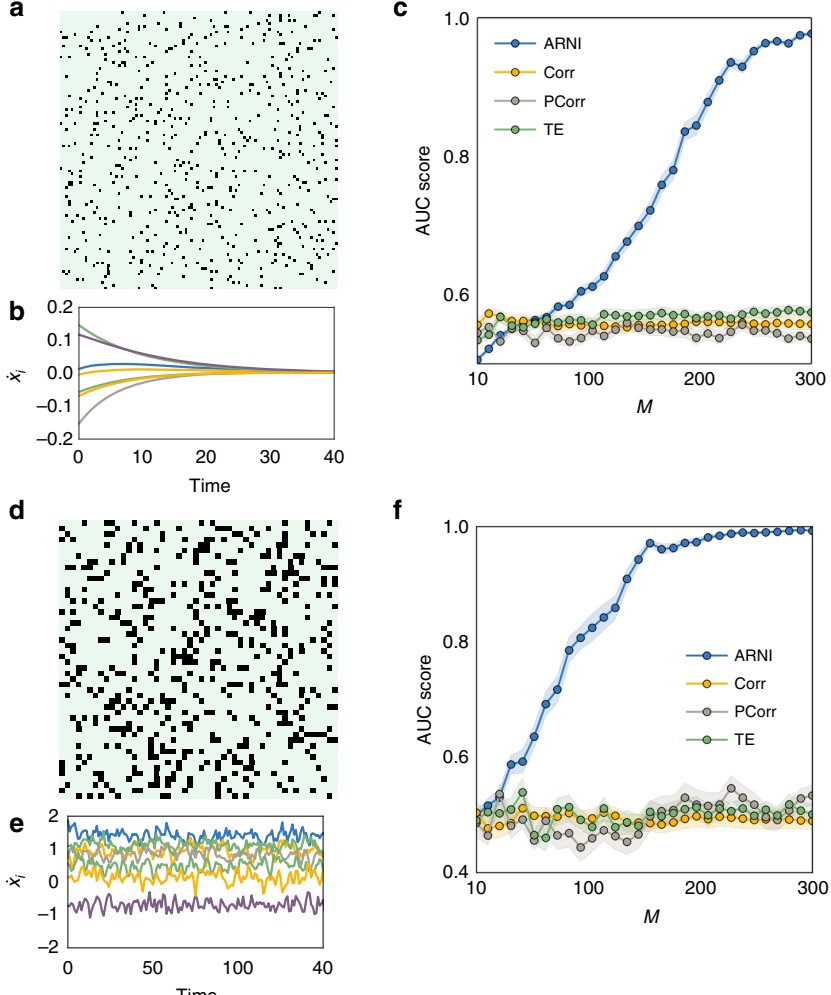

**Fig. 2** Inferring interactions from transients to steady state and from non-periodic dynamics. We simulated short transient time series of $m = 5$ time points starting from different initial conditions and created composite longer time series as in Fig. 1. Thus $M = S \times m$, where $S$ is the number of distinct transient dynamics. **a–c** Revealing interactions from transients toward steady state. **a** Adjacency matrix of a network of $N = 100$ units under Michaelis–Menten kinetics with $n_i = 10$ incoming connections. **b** Example of transient dynamics toward steady states, where $\dot{x}_j = 0$ for all $j \in \{1, 2, ..., N\}$. **c** Quality of inferences with respect to $M$ using our approach (ARNI), correlations (Corr), partial correlations (PCorr), and transfer entropy (TE). **d–f** Revealing interactions from non-periodic dynamics. **d** Adjacency matrix of a network of $N = 50$ phase-coupled oscillators with $n_i = 10$ incoming connections (black squares) per unit. **e** Example of derivatives for several oscillators. **f** Quality of reconstruction from short trajectories with respect to $M = S \times m$ with $m = 10$

appropriate for reconstruction, see, e.g., Fig. 1c. Exemplary tests demonstrate that even time series as short as $m = 5$ time points recorded from dynamics from different trajectories evolving toward a steady state might be sufficient. Moreover, reconstruction quality improves with the total number of available recordings $M = S \times m$ where $S$ is the number of experiments, in contrast to inferences from thresholding correlations, partial correlations, and transfer entropy, which cannot predict existing interactions under these minimal sampling conditions (Fig. 2a–c). Moreover, inference studies on collections of short time series extracted from non-periodic dynamics further confirms that larger numbers $M = S \times m$ of recordings improve quality (as expected). Again, correlations, partial correlations, and transfer entropy are in general less capable of capturing the intrinsic structure of interactions under equally minimal conditions (Fig. 2d–f). Finally, interactions may still be recovered in networks of higher-dimensional units by extending Eq. (5) to include all components $x_i^d(t)$ of $i \in \{1, 2, ..., N\}$, where $d \in \{1, 2, ..., D_i\}$ and $D_i$ is the number of components of unit $i$, Fig. 3a–c.

**Performance**. To further characterize the performance of our approach, we carried out systematic reconstructions of various networks of different sizes, numbers of incoming connections per unit, noise levels, fraction of higher-order (hypernetwork) inter-actions, and number of hidden units (Fig. 4). We report four classes of results. First, the number $M_\theta$ of time points necessary for AUC scores larger than a threshold $\theta$ scales sublinearly with the size of the network, Fig. 4a, and linearly with the number of incoming connections per unit, Fig. 4b. Moreover, inferring the incoming connections of single units in large sparse networks ($N = 1000$, $n_i = 10$) in conventional hardware (Intel® Core™ i5-2430M) takes $65 \pm 26$ s per unit. Such results highlight the potential applicability of our approach in combination with parallel computing for revealing interactions in real-world networks, which are often large in size and sparsely connected. Second, $M_\theta$ depends supralinearly on the noise level $\eta$, Fig. 4c. Here, sampling longer time series (more data) improves recon-struction quality. These results indicate that inference is still viable for highly noisy dynamics at the expense of recording longer time series. Third, systematic reconstructions of

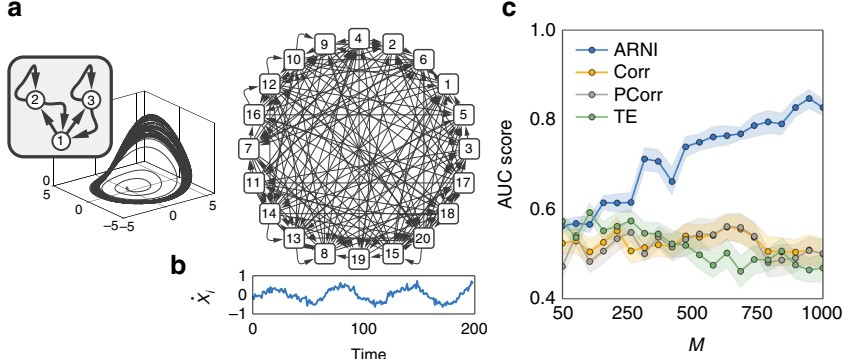

**Fig. 3** Inferring interactions from noisy and chaotic dynamics. Reconstruction of chaotic oscillator networks of $N = 20$ and $n_i = 5$ incoming interactions. **a** Rössler oscillators themselves constitute subnetworks of three interconnected units. **b** Example of noisy derivatives in a Rössler oscillator. **c** Quality of reconstruction from transient trajectories with respect to $M = S \times m$ with $m = 10$, using our approach (ARNI), correlations (Corr), partial correlations (PCorr), and transfer entropy (TE)

hypernetwork interactions in exemplary models of phase-coupled oscillators (Supplementary Note 4) suggest that our results are independent of the probability of having hypernetwork interactions $p_h$, Fig. 4d, e. This is a consequence of treating pairwise and higher-order interactions equally, by decomposing the coupling into orders of jointly acting units via explicit dependency matrices (Eq. (3)). Thus the approach is insensitive to the appearance of higher-order interactions. Finally, even if some units of the network are not measured (hidden units), existing and non-existing links among measured units may still be reliably inferred, Fig. 4f. To compute AUC scores, we compare our predictions for the existence and absence of links among the measured units with those actually existing and not existing among those units, making no statement about indirect interactions mediated by hidden units. As more units are hidden, the quality of reconstruction decreases because the hidden units act upon the measured units in an unknown way. Still, sampling longer time series again improves reconstruction quality. Thereby, the model-free approach provides accurate predictions even if only a fraction of the network is recorded.

**Proper basis functions and learning curves.** Selecting an appropriate class of basis functions to represent the network interactions in system (Eq. (6)) is vital for any such approach. Choosing basis functions that capture the intrinsic nature of interactions (e.g., $h(x_i)$, $h(x_i, x_j)$, $h(x_i, x_j, x_w)$, and so on) by construction yields optimal results. However, to exactly pick the correct interaction function requires prior knowledge of the potential functions involved in coupling units of the system under consideration. To overcome this limitation, we aim at appropriate classes of coupling functions only but do not require to pick a correct function (that would enable prediction of time series). While the former implies to find basis functions of correct order, the latter implies to find a unique set of basis functions capable of fitting the recorded dynamics (see below for further consequences). We remark that a particularly chosen basis function constitutes a representative of an entire class of appropriate functions. For instance, the functions indexed $a$–$d$ in Table 1 are all equally appropriate representatives of the class of pairwise functions $g_{ij}^i(x_i, x_j)$, Fig. 5.

We investigated the effects of selecting different basis functions. For the example shown in Fig. 5, we studied networks of phase-coupled oscillators and divided the time series in a training set (60% of time points) for inferring interactions and a validation set (40% of time points) for evaluating the predictions;

we tracked the evolution of a fitting cost function with respect to the $l$-th discovered interaction. Specifically, the fitting cost function is defined as

$$C_i(l) := \frac{1}{M_s} \sum_{m=1}^{M_s} \left( \dot{x}_{i,m} - \hat{\dot{x}}_{i,m}(l) \right)^2, \tag{7}$$

where $M_s$ is the number of time points in the set and $\hat{\dot{x}}_{i,m}(l) \in \mathbb{R}$ is the prediction by our approach of a computed $\dot{x}_{i,m}$ using the inferred interactions up to the $l$-th discovered interaction.

The functional forms of the cost function $C_i(l)$, depending on the number $l$ of interactions considered, are either L-shaped, indicating the number of incoming connections at the knee $l^*$ of the L (basis functions $a$–$d$ of Table 1, Fig. 5a–d), or not, thereby not revealing any features of the network (basis functions $e$ and $f$ of Table 1, Fig. 5e, f). Simultaneously to reveal the number of incoming connections, the first $l^*$ interactions actually chosen provide the full information about which units $j$ directly act on unit $i$. We remark that, for sufficiently short sampling intervals, both the time derivative $\dot{x}_{i,m}$ as well as its estimator $\hat{\dot{x}}_{i,m}$ are obtainable from recorded dynamics data without any model assumption.

These findings confirm that basis functions that merely capture the essential structure of the interactions but not necessarily exactly represent the full dynamics are sufficient to reveal network connectivity. As a consequence, reconstruction of direct network interactions is possible without preknowledgde about a system model.

**Effects of noise and hidden units.** In experimentally relevant biological settings, there may be several uncontrolled factors affecting the recorded time series. For instance, in gene networks, noisy dynamics is simultaneously present at several different levels (e.g., gene-intrinsic, network intrinsic, and cell-intrinsic)[2]. Fundamentally, noise complicates the inference process by corrupting measurements of units dynamics, thereby masking network interactions. Moreover, one may not have complete access to measure all units in the network. This may induce correlating or decorrelating effects among units, thus promoting the recovery of indirect interactions[34,47].

To test the robustness of our approach against the combination of both noise and hidden units, we simulated transients toward steady states under the external influence of Gaussian noise and recorded the dynamics of only a subset of randomly selected units in the network. Results indicate that both noise and hidden units

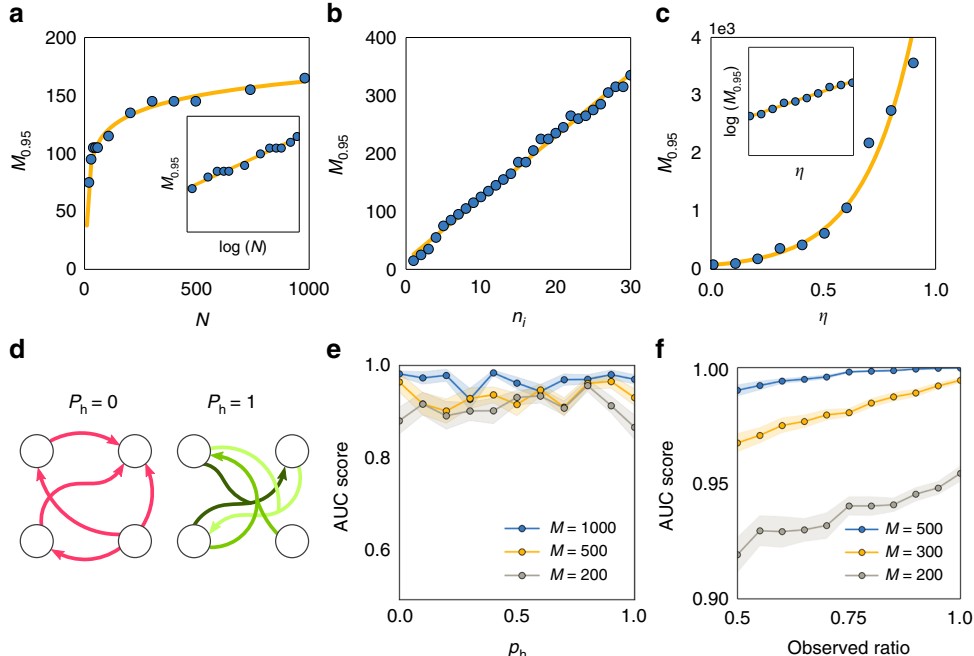

**Fig. 4** Performance on networks and hypernetworks. **a** Minimum length $M_{0.95}$ of time series required for achieving AUC score >0.95 versus the number of units $N$ with a logarithmic fit. The inset shows the same data with $N$ on logarithmic scale. The number of incoming interactions $n_i = 10$ was fixed for all networks. **b** $M_{0.95}$ versus $n_i$ with a linear fit, for networks of fixed size $N = 50$. **c** $M_{0.95}$ versus standard deviation of noise level $\eta$ with a exponential fit. The inset shows the same data with $M_{0.95}$ on logarithmic scale. **d** Cartoon of networks with $p_h = 0$ and $p_h = 1$. **e** Systematic reconstructions of hypernetworks of $N = 20$ phase-coupled oscillators for varying $p_h \in [0, 1]$ and $M \in \{200, 500, 1000\}$ and fixed $n_i = 5$. **f** Systematic reconstructions of networks of $N = 20$ ($n_i = 10$) units versus the fraction of observed (non-hidden) units

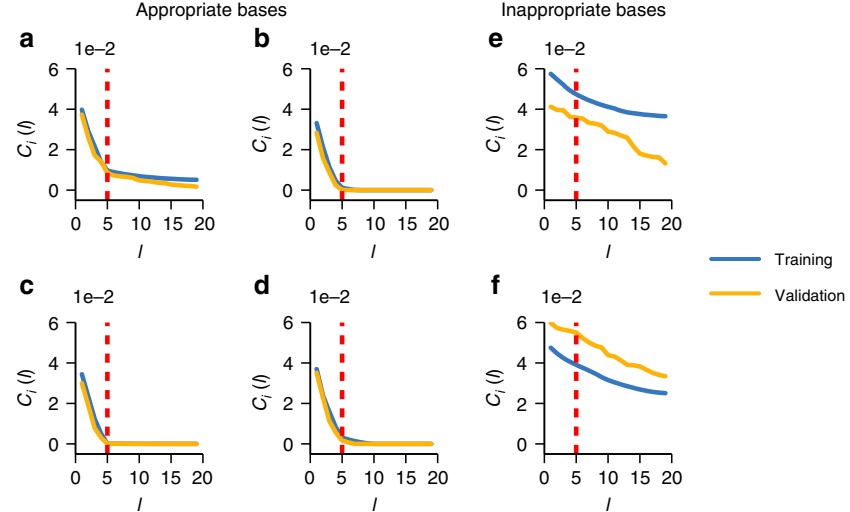

**Fig. 5** Learning curves with respect to the number of iterations $l$ reveal appropriate expansions. **a–f** Learning curves for basis functions shown in Table 1; indices in Table 1 equal panel identification. The training and validation sets are composed by the 60 and 40% of recordings, respectively. For all basis functions **a–d** chosen from an appropriate class (here: true pairwise interactions), the curve exhibits an L-shape with a clear plateau starting at the correct number of incoming interactions (vertical dashed line), above adding more interactions reduces the fitting costs only weakly or not at all. If basis functions are chosen from an inappropriate class (**e**, **f**), the cost functions decrease only mildly and do not exhibit L-shape

moderately reduce the performance of our approach, Fig. 6. However, the inference quality still increases with $M$, Fig. 6d, such that larger sampling collections may still reveal interaction topology. Moreover, systematic reconstructions of different sets of recorded units indicate that our predictions generally outperform those extracted from correlations, partial correlations, and transfer entropy.

**Robust inference of biological networks**. Next we establish the potential of our framework to reconstruct interactions for biological system settings. Specifically, we demonstrate results on two networked model biological systems: glycolytic oscillator in yeast[48] and circadian clock in *Drosophila*[49]. The glycolytic oscillator, exhibiting one of the classical examples for cellular oscillations, accounts for the main reactions of glycolysis. Here we

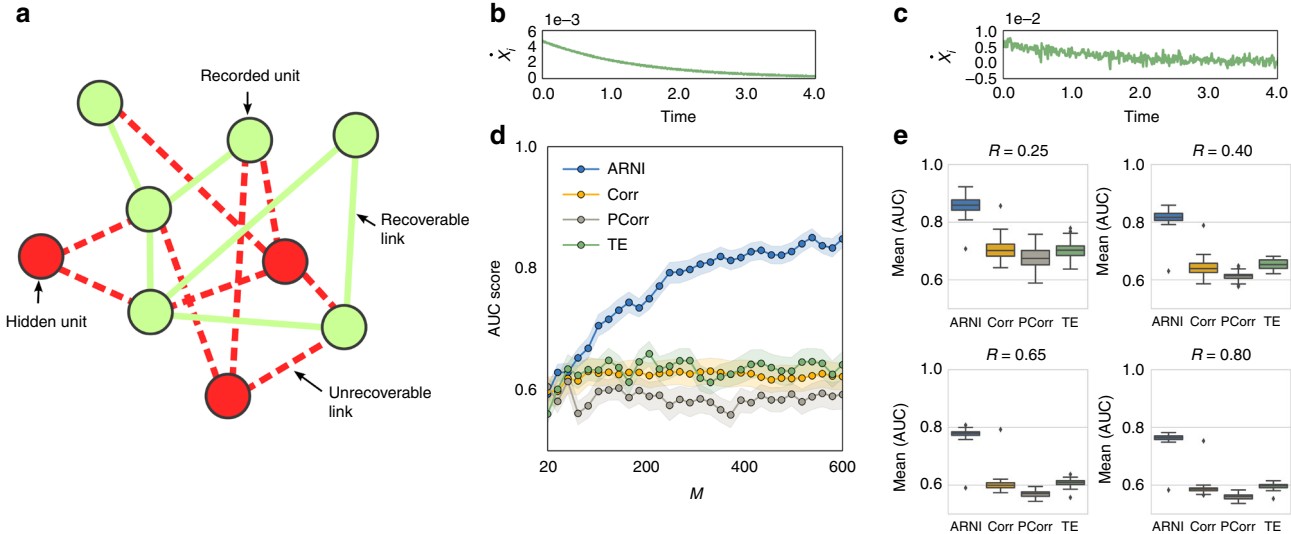

**Fig. 6** Reconstructions are still viable if units are hidden and measurements are noisy. Revealing direct interactions within a subset of units from noisy transients toward steady states. The network size is fixed at $N = 100$. **a** Representation of a network with a subset of measured units (green) and a subset of hidden units (red). **b, c** $\dot{x}_i$ in noise-free and noisy transients. **d** Quality of reconstruction versus number of measurements employing our approach (ARNI), correlations (Corr), partial correlations (PCorr), and transfer entropy (TE) on a subset of 40 (randomly selected) recorded units. **e** Systematic reconstruction over different collections of subsets. The variable $R$ indicates the fraction of recorded units. Averages over 50 random subsets of $R < 1$ indicate that our approach outperforms correlations, partial correlations, and transfer entropy across different $R$ values

## Table 1 Interactions may be represented in different basis functions

| Index | Basis function |
|-------|----------------|
| a | $h^i_{ij,p}(x_i, x_j) = (x_j - x_i)^p$ |
| b | $h^i_{ij,p_1 p_2}(x_i, x_j) = x_i^{p_1} x_j^{p_2}$ |
| c | $h^i_{ij,p}(x_i, x_j) = sin(p(x_j - x_i))$ and $h^i_{ij,p}(x_i, x_j) = cos(p(x_j - x_i))$ |
| d | $h^i_{ij,p}(x_i, x_j) = 1 + \left\| \boldsymbol{x}_{ij} - \boldsymbol{x}_{ij,p} \right\|^2$, $\boldsymbol{x}_{ij} = (x_i, x_j)$ and $\boldsymbol{x}_{ij,p} = (x_{i,p}, x_{j,p})$ |
| e | $h^i_{j,p}(x_j) = x_j^p$ |
| f | $h^i_{j,p}(x_j) = sin(px_j)$ and $h^i_{j,p}(x_j) = cos(px_j)$ |

Different basis functions can be employed to represent a specific interaction. Basis functions of correct order are more likely to reveal true interactions than functions of incorrect order (see also Supplementary Note 6), Fig. 5. Basis function *d* belongs to a class of radial basis functions, so $\boldsymbol{x}_{ij,p}$ represent the *p*-th center of the expansion[56]

focus on a model for anaerobic glycolytic oscillations in yeast, containing the influx of glucose and outflux of pyruvate and/or acetaldehyde[48] (see Supplementary Note 5 for an extended description). The circadian clock underlies the biological response to the day–night cycle, and the oscillations it exhibits in *Drosophila* are driven by a negative feedback between two genes and the complex that is formed by the proteins they code for. The model equations for the circadian clock are based on ref. [49] (see Supplementary Note 5).

Employing the above approach of combining a dynamics space representation, expanding in suitable families of basis functions, and solving the resulting linear regression problem by an orthogonal least squares method, we reconstructed the interactions between the different components of the glycolytic oscillator (Fig. 7a, b) and the circadian clock (Fig. 7c, d) from transient dynamics toward their periodic orbits. As for the other systems' settings, the results confirm that larger number $M$ of observations improve the predictions. Moreover, the reconstruction quality by this method again outperforms those resulting from correlations, partial correlations, and transfer entropy.

## Discussion

We proposed a model-free framework for inferring direct interaction networks from only the time series of collective nonlinear system dynamics. First, defining the notion of explicit dependency matrices enabled us to systematically decompose each units' dynamics into pairwise, three-point, and higher-order interactions and at the same time treat present influences from one unit to another on the same footing independently of the interaction order. Second, by capturing the structure (but not necessarily the exact functional form) of the dynamical influences through appropriately chosen basis functions, we posed the reconstruction problem based on nonlinear dynamics as a mathematical regression problem with grouped variables. Given that the reconstructions of the sets of incoming connections to different units of the network are mathematically independent (despite using overlapping recorded dynamical data), the framework is scalable (see Supplementary Note 2) and computationally parallelizable for large networks. Reconstruction is robust across a wide range of dynamical regimes, combined pairwise and hypernetwork interactions, noise, and hidden units.

The main advantage of our framework is its minimal sampling conditions. For instance, in systems during transients to steady states (such as in gene regulation[3,5,7]) or periodic orbits (such as in glycolytic oscillations[48]), we reconstructed direct interactions without the need to know the actual strength or actual distributed patterns of perturbations from those states. In contrast to several previous studies[1,11,13,49], our framework in general does not require to apply external driving signals and if a system is externally driven, e.g., to create transients, these signals need not be controlled; thus our framework might be suitable for systems not easily accessible for controlled driving or external driving at all. Moreover, collections of very short time series, in practice potentially resulting from different experiments on the same system, are sufficient for reconstruction. In particular, collective dynamics that is transient, stochastically driven, or otherwise sufficiently complex helps revealing interactions, whereas certain stable dynamics on low-dimensional subsets of state space only sample limited regions of the dynamics space and thus in

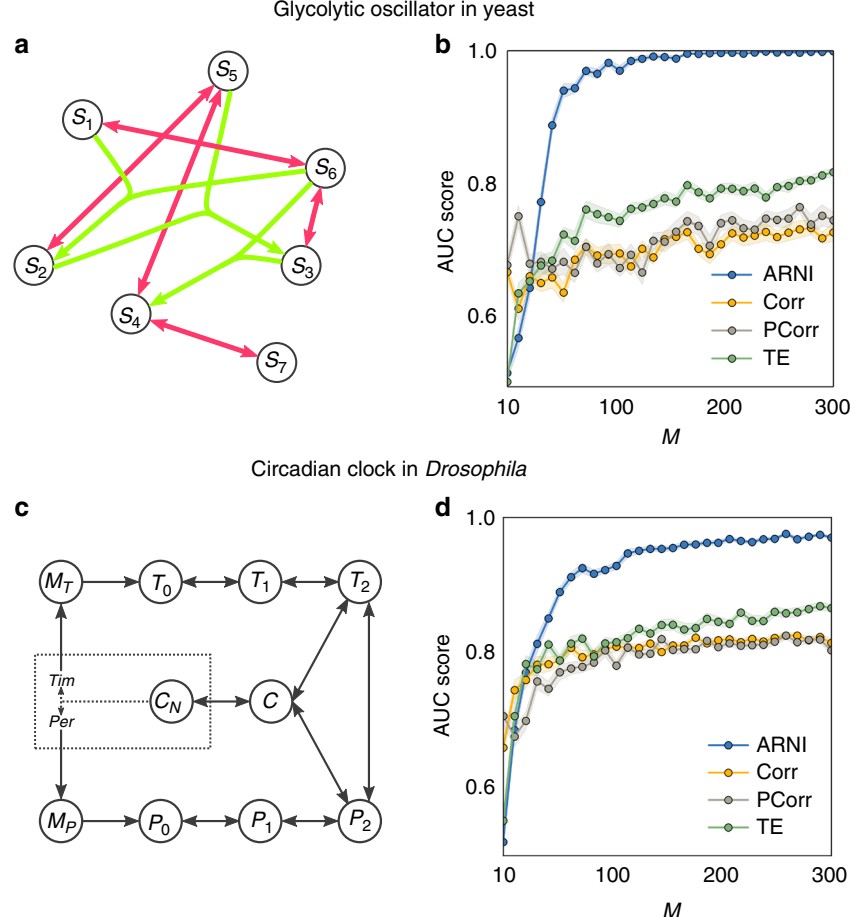

**Fig. 7** Reconstructing biological model systems. **a–c** Revealing interactions of glycolytic oscillator in yeast. **a** Glycolytic oscillator network, red and green links represent network and hypernetwork interactions, respectively. **b** Quality of reconstruction from transient trajectories as a function of $M = S \times m$ with $m = 10$. **c** Circadian clock network in *Drosophila*. **d** Quality of reconstruction as in **b**

principle do not provide full information about network interactions. Lower-dimensional dynamics may in particular be induced by symmetries or other invariants represented by algebraic conditions, such as $z(x) = 0$. For instance, in systems evolving in synchronized states, the existence and directionality of interactions are impossible to extract from time series[32]. Furthermore, the number of independent measurements required for successful reconstruction grows linearly with the local number of interaction partners and sublinearly with the number of units in the network, providing an advantage for reconstructing large systems. As we illustrated by examples, our framework may be easily combined with learning curves derivable from recorded data only and thus enables researchers to determine the accuracy of inferences when there is no ground truth available.

Previous studies on inferring the direct interaction structure from time series have focused on the reconstruction of networks with known local dynamics and coupling functions[11,12,14–18]. Such prior knowledge reduces the task to a standard linear algebra problem, where one has to solve linear systems of equations to reveal the network connections, cf. ref.[11] for a comprehensive review. Recent work on low-dimensional dynamical systems[20], based on expanding the system dynamics in basis functions, requires the dynamics to admit a sparse representation in the proposed basis. Moreover, a work[21] applying an extension of the method described in ref.[20] on models for gene regulation also suggests that such approaches scale supralinearly with the dimensionality of the network for both the number of candidate coupling functions and the time points necessary for successful

reconstruction. The theory presented above does neither require prior knowledge of parameters and coupling functions involved in the network dynamics nor does it require these functions to admit a sparse representations in any basis chosen; it is not limited to low-dimensional networked systems, also because the number of necessary time points for successful reconstruction scales sublinearly with network size.

Taken together, this model-free, robust framework can be based on collections of short time series, noisy data, partially inaccessible units, and essentially arbitrary nonlinear dynamics and may thus enable the reconstruction of direct interaction networks from dynamical data from a new range of times series from coupled dynamical systems where no model is known.

## Methods

**Overview**. To generate dynamical trajectories displaying transients toward steady states, we simulated dynamical systems employing Michaelis–Menten kinetics (Supplementary Note 4), systems frequently used to model gene regulation[3,5,27]. To generate dynamical trajectories exhibiting transients to periodic dynamics, we employed two biological model systems: (i) glycolytic oscillations in yeast[48] and circadian clock in *Drosophila*[49] (Supplementary Note 5), which possess hypernetwork interactions, where two units jointly and directly influence a third such that their interaction function cannot be disentangled into sums of pairwise interactions. To study the effects of non-periodicity, we simulated networks of phase-coupled oscillators (Supplementary Note 4) whose coupling stems from a simple model of weakly coupled populations of biological neurons[51–54]. Finally, to test robustness against chaos and noise, we simulated networks of noisy and asynchronous Rössler oscillators (Supplementary Note 4), prototypical systems for studying chaos[55].

In what follows, we provide a brief description of each model (see Supplementary Notes 4 and 5 for further details).

**Gene regulatory circuits.** To simulate systems mimicking gene regulation, we simulated networks of dynamical systems having Michaelis–Menten kinetics[3,27]

$$\dot{x}_i = -x_i + \frac{1}{n_i}\sum_{j=1}^{N} J_{ij}\frac{x_j}{1+x_j} + \xi_i, \tag{8}$$

having $n_i$ randomly-selected incoming connections per node. Here $J_{ij}$ of $J \in \mathbb{R}^{N\times N}$ represents a weighted and directed link from unit $j$ to $i$.

**Networks and hypernetworks of phase-coupled oscillators.** To generate non-periodic dynamics, we simulated a model[52] of phase-coupled oscillators with coupling functions having two Fourier modes

$$\dot{x}_i = \omega_i + \frac{1}{n_i}\sum_{j=1}^{N} J_{ij}\big[\sin(x_j - x_i - 1.05) + 0.33\sin(2(x_j - x_i))\big] + \xi_i, \tag{9}$$

with constant natural frequencies $\omega_i$.

We extended this model to hypernetworks of the form

$$\dot{x}_i = \omega_i + \frac{1}{n_i}\sum_{j=1}^{N}\sum_{k=1}^{N} E^i_{jk}\big[\sin(x_j - x_k - 1.05) + 0.33\sin(2(x_j - x_k))\big] + \xi_i. \tag{10}$$

Differently from (Eq. (9)), here we introduce the second-order interaction matrix $E^i \in \mathbb{R}^{N\times N}$ for all $i = \{1, 2, \ldots, N\}$. Specifically, the element $E^i_{jk}$ quantify how strongly units $j$ and $k$ jointly and directly influence unit $i$.

**Networks of Rössler oscillators.** To generate chaotic dynamics, we simulated networks of coupled Rössler oscillators[55]. The dynamics of each oscillator $\boldsymbol{x}_i = \big[x_i^1, x_i^2, x_i^3\big] \in \mathbb{R}^3$ is set by three differential equations

$$\dot{x}_i^1 = -x_i^2 - x_i^3 + \frac{1}{n_i}\sum_{j=1}^{N} J_{ij}\sin(x_j^1) + \xi_i^1, \tag{11}$$

$$\dot{x}_i^2 = x_i^1 + 0.1 x_i^2 + \xi_i^2, \tag{12}$$

$$\dot{x}_i^3 = 0.1 + x_i^3(x_i^1 - 18.0) + \xi_i^3, \tag{13}$$

where $\xi_i^k$ with $k \in \{1, 2, 3\}$ represent external noisy signals acting on the unit's components.

**Glycolytic oscillator model.** To test performance on biological model systems, we first simulated the glycolytic oscillator defined as[48]

$$\dot{S}_1 = J_0 - \frac{k_1 S_1 S_6}{1 + (S_6/K_1)^q} \tag{14}$$

$$\dot{S}_2 = 2\frac{k_1 S_1 S_6}{1 + (S_6/K_1)^q} - k_2 S_2(N - S_5) - k_6 S_2 S_5 \tag{15}$$

$$\dot{S}_3 = k_2 S_2(N - S_5) - k_3 S_3(A - S_6) \tag{16}$$

$$\dot{S}_4 = k_3 S_3(A - S_6) - k_4 S_4 S_5 - \kappa(S_4 - S_7) \tag{17}$$

$$\dot{S}_5 = k_2 S_2(N - S_5) - k_4 S_4 S_5 - k_6 S_2 S_5 \tag{18}$$

$$\dot{S}_6 = -2\frac{k_1 S_1 S_6}{1 + (S_6/K_1)^q} + 2k_3 S_3(A - S_6) - k_5 S_6 \tag{19}$$

$$\dot{S}_7 = \psi\kappa(S_4 - S_7) - k S_7 \tag{20}$$

where $S_1$ represents the concentration of glucose, $S_2$ that of glyceraldehydes-3-phosphate and dihydroxyacetone phosphate pool, $S_3$ that of 1, 3-bisphosphoglycerate, $S_4$ that of cytosolic pyruvate and acetaldehyde pool, $S_5$ that of NADH, $S_6$ that of ATP, and $S_7$ that of extracellular pyruvate and the acetaldehyde pool.

**Circadian clock.** A second biological model system we have studied is the circadian clock, underlying the response to the day–night cycle. It is defined as[49]:

$$\dot{M}_P = v_{sP}\frac{K_{IP}^n}{K_{IP}^n + C_N^n} - v_{mP}\frac{M_P}{K_{mP} + M_P} - k_d M_P \tag{21}$$

$$\dot{P}_0 = k_{sP}M_P - V_{1p}\frac{P_0}{K_{1P} + P_0} + V_{2p}\frac{P_1}{K_{2P} + P_1} - k_d P_0 \tag{22}$$

$$\dot{P}_1 = V_{1p}\frac{P_0}{K_{1P} + P_0} - V_{2p}\frac{P_1}{K_{2P} + P_1} - V_{3p}\frac{P_1}{K_{3P} + P_1} + V_{4p}\frac{P_2}{K_{4P} + P_2} - k_d P_1 \tag{23}$$

$$\dot{P}_2 = V_{3P}\frac{P_1}{K_{3P} + P_1} - V_{4p}\frac{P_2}{K_{4P} + P_2} - k_3 P_2 T_2 + k_4 C - v_{dP}\frac{P_2}{K_{dP} + P_2} - k_d P_2 \tag{24}$$

$$\dot{M}_T = v_{sT}\frac{K_{IT}^n}{K_{IT}^n + C_N^n} - v_{mT}\frac{M_T}{K_{mT} + M_T} - k_d M_T \tag{25}$$

$$\dot{T}_0 = k_{sT}M_T - V_{1T}\frac{T_0}{K_{1T} + T_0} + V_{2T}\frac{T_1}{K_{2T} + T_1} - k_d T_o \tag{26}$$

$$\dot{T}_1 = V_{1T}\frac{T_0}{K_{1T} + T_0} - V_{2T}\frac{T_1}{K_{2T} + T_1} - V_{3T}\frac{T_1}{K_{3T} + T_1} + V_{4T}\frac{T_2}{K_{4T} + T_2} - k_d T_1 \tag{27}$$

$$\dot{T}_2 = V_{3T}\frac{T_1}{K_{3T} + T_1} - V_{4T}\frac{T_2}{K_{4T} + T_2} - k_3 P_2 T_2 + k_4 C - V_{dT}\frac{T_2}{K_{dT} + T_2} - k_d T_2 \tag{28}$$

$$\dot{C} = k_3 P_2 T_2 - k_4 C - k_1 C - k_2 C_N - k_{dC} C \tag{29}$$

$$\dot{C}_N = k_1 C - k_2 C_N - k_{dN} C_N \tag{30}$$

where $M_T$ and $M_P$ are *tim* and *per* mRNAs, respectively. $T_0$, $T_1$, and $T_2$ are forms of the TIM protein, $P_0$, $P_1$, and $P_2$ are forms of the PER protein, and $C$ and $C_N$ are forms of the PER–TIM complex.

**Data availability.** All data reported in this study are available from the corresponding authors upon request. Example codes for simulating and reconstructing network dynamical systems may be found at https://github.com/networkinference/ARNI.

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

## Acknowledgements

We thank Fabio Schittler Neves, Benedict Lünsmann and Fenna Müller for useful discussions. M.T. thanks Albert Laszlo Barabasi for hospitality and useful discussions during a visit in March 2016. We acknowledge support by the German Research Foundation and the Open Access Publication Funds of the TU Dresden. This work is supported through the German Science Foundation (DFG) by a grant toward the Center of Excellence "Center for Advancing Electronics Dresden" (cfaed). We also gratefully acknowledge support from the Federal Ministry of Education and Research (BMBF Grant Nos. 03SF0472E and 03SF0472F) and the Max Planck Society.

## Author contributions

All authors conceived the research and contributed materials and analysis tools. J.C. and M.T. developed the theory and algorithms and designed the research. All authors provided model systems and the quality measures. J.C., M.N., and S.H. carried out the numerical experiments. All authors analyzed the data, discussed and interpreted the results, and wrote the manuscript.

## Additional information

**Competing interests:** The authors declare no competing financial interests.

