## [Peer Review File · Nature Communications]

Reviewers' comments:

Reviewer #1 (Remarks to the Author):

Model-free inference of direct interactions from nonlinear collective dynamics

Casadiego, Nitzan, Hallerberg, Timme

This paper explores sparse methods to identify networks of dynamical interactions. The authors use group sparse regression in a framework that identifies higher order interactions. They also demonstrate the method on a number of examples, investigating the learning rate and comparing with other leading methods.

The paper is interesting, and the analyses and figures are clear and compelling. Identifying network structure with sparse methods is a good idea for the important field of network inference. However, there are a number of important issues that need clarification before I can recommend publication.

* It is important to clarify the key differences between the present work and the authors' recent Science Advances paper. They seem extremely close in theory... both identify networks using compressed sensing? This is important to clarify to assess if the present work is sufficiently novel.

* Can you clarify: does this method identify the graph of connections but not the specific nonlinear terms in the dynamics? Can it identify the terms once the graph structure is identified?

* I believe that the code to generate and analyze data should be included along with the manuscript for the referees and for the readers. This will make the method much more transparent, and will promote reproducible research.

* How does the method perform if the system is sampled without any transients. I can imagine many systems where the dynamics/interactions could be ambiguous on a limit cycle, but where in fact there are hidden variables and connections that only appear with transients. This might be clarified in the paper.

* This work seems quite closely connected to related methods that identify dynamics with sparse methods, such as references [14,32]. You mention these methods on page 6, and only in passing to say that this method is fundamentally different because it doesn't require exact representation in a basis or need to admit a sparse representation. However, it really doesn't look very different, as it relies on the choice of a basis "h" and then uses sparsity to identify connections. I am not saying that it is the same, but to brush over the strong similarities seems like a mistake. In fact, this does not strengthen the impact of the present work, but isolates it. Also, Reference [14] is mischaracterized in the introduction (line 34): this method does not only determine unknown parameters, but identifies underlying connections and nonlinear dynamics.

* A recent paper by Mangan et al (IEEE Transactions on Molecular, Biological... "Inferring biological networks by sparse identification...") uses related sparse methods to identify networks for biological models. These methods identify Michaelis Menten kinetics and yeast glycolysis networks (both methods that you identify/mention). This is an important reference to discuss/compare.

* There are statements in the discussion about how this method scales better to large networks. Without including code or a computational complexity analysis, it is unclear if the proposed method actually scales to larger systems better than alternative methods.

* How does this method scale to high dimensional systems? What is the highest dimension tested?

* How would this method work if the dynamical interactions are more complex (sinusoidal, or other generalized functions)? What if the dynamics functions have no support in the basis you choose for "h"? Is the choice of basis explored, and multiple bases compared?

Typos:

- After equation 1, comma missing before "x_N"

- Same paragraph: Generally noise added to the state of a differential equation is referred to as a "disturbance" (as opposed to noise added to measurements that do not affect the dynamics of the system).

Reviewer #2 (Remarks to the Author):

The authors propose a new method for inferring direct interactions from observed time series generated from coupled nonlinear systems. The key idea for the proposed method is to consider the superposition of general functions with an argument, two arguments, three arguments and forth to represent the underlying dynamics. Then, the authors fit the superposition by minimizing a projection error function. The performance for the proposed method outperforms those for correlations, partial correlations and transfer entropy. In addition, the results presented with the AUC are in the highest standards I have seen among the papers on network reconstructions. Thus, researches following this work should regard this work as a role model. Therefore, practically, this work is appealing and should be published.

However, there are some concerns I would like to raise:

(i) The authors do not declare their assumptions clearly, one of which is that a time series is generated by transient states or stochastic dynamics. Thus, the authors should emphasize such assumptions. If they also want to include a set of nonlinear deterministic systems on the attractors, they need to (a) cite more appropriate papers including Nawrath et al., *Phys. Rev. Lett.* (2010) and Zou et al., *Int. J. Bifurcat. Chaos* (2011), and (b) comment on the effect of synchronization (Paluš & Vejmelka, *Phys. Rev. E*, 2007) as well as the effect of hidden common causes (Hirata & Aihara, *Phys. Rev. E*, 2010).

(ii) Their statement that one can represent uniquely the dynamics in the right hand side of Eq. (4) does not always seem to be valid. According to the paper by Judd and Nakamura, *Chaos* (2006), one cannot necessarily choose the correct model as the best model and some equivalent models exist. Even if we do not use any delays, simply imagine the case where there is some invariant equation represented by $z(x) = 0$. Then, after solving this invariant equation in terms of x if necessary, we can generate various equivalent equations. Thus the representations on the right hand side of Eq. (4) will become non-unique, and the “best” model might have an extra mechanism of noise reduction. Thus, the authors need to prove or assume that such an invariant equation does not exist for dynamics they analyze to ensure the uniqueness of the representations. This is another reason why the authors should emphasize their assumptions more.

(iii) In Figure 3f, define more detail on how they define the AUC score when there are hidden elements. The score of more than 90% are too large if the half of the elements are hidden because then we cannot access about 75% of pairwise connections.

(iv) The references for partial correlations and transfer entropy are not shown appropriately. In addition, I had an impression that correlations, partial correlations, and transfer entropy the authors chose to compare are too old and not in the current state of the art. This point is obvious because they cited many recent methods but did not compare the proposed method with them quantitatively. Thus, if time allows, the authors might want to revise this point so that we can

accelerate the researches in this field and enhance the discussions. But, this latter part is optional for the revision.

(v) Although the authors state in the main text and the title that their method is model-free, their performance depends on which basis functions they use (see Fig. S5). Thus, the title of “model-free inference” might be overstated and should be rephrased appropriately.

Although I have these concerns, I found that the manuscript should be published after some minor revision because their results will attract attention of wide audience, and their way of evaluation using the AUC should become our future standard practice in this field.

Reviewer #3 (Remarks to the Author):

In this paper, the authors propose a model-free method for inferring interactions of system elements (or units) of the 2nd or even higher order. The problem is definitely important in the study of non-linear dynamic system.

Given a non-linear dynamic system

$$d(x_i)/d(t)=F_i(x(t))+\eta(t),$$

the authors determine the interaction relationships by decomposing $F_i(x(t))$ into a (finite) series at the given time points:

$$d(x_i)/d(t)=\sum_j \Lambda^{i}_{jj}g^{i}_{j}(x_j) + \sum_{\{j, s\}} \Lambda^{i}_{jj} \Lambda^{i}_{ss} g^{i}_{\{j, s\}}(x_j, x_s) + \dots$$

where $g^{i}_{j}()$, $g^{i}_{\{j, s\}}()$ are a LINEAR COMBINATIONS of some basis functions (listed in table 1, for example).

Additionally, the authors give a method for such a decomposition.

Studying the properties of a function through decomposition is not new, for which an exemplar is Fourier analysis or wavelet.

The authors do not answer the following questions regarding their method well:

(a) Why are the basis functions in Table 1 used?

(b) Are the decomposition with respect to the basis functions in Table 1 used in wavelet theory, or somewhere else?

(c) is there any specific real application that can be used to demonstrate the usefulness of the proposed method? for example, in inference of gene regulatory relationship, which is mentioned several times in the manuscript.

Minor comments.

The manuscript is prepared poorly. There are many grammar mistakes, ambiguous concept and awkward sentences. Some figures are not understandable. Here is the list of some instances.

1. Line 17, page 2. "amount" not "amounts".

2. Line 39, page 2. units are not defined? what are units?

3. Line 80, page 3. in (3), λ_{ij} has unnecessary DOUBLE subscript. λ_{ij} is enough.

4. Line 99, page 5. In formula (5), the term $\eta_i(t)$ is mysteriously missing from formula (4).

5. Line 105, page 6. punctuation problem before "as"

6. Figure 2 is positioned correctly. Part (b) is not understandable. "a relaxation to a steady state" means what?

7. Figure 5. The used network has 100 units. Why the figure (b) and (c) were drawn for 40 units. how these 40 units were selected?

8. Page 5, Supplementary document. Step 1 is "Select a generic model for interactions by choosing orders K, P_1, P_2, P_3 and so on, in (5)". P_1, P_2, P_3 are not found in (5) on page 5.

GENERAL COMMENTS BY THE REVIEWERS

We are grateful to the reviewers for carefully reviewing our work. Overall, the reviewers evaluated the topic addressed as well as the approach proposed and the results presented in a positive way. At the same time, they have specific questions, comments or suggestions to modify and improve the manuscript.

Specifically, Reviewer 1 finds that the work is “*interesting*”, “*the analyses and Figures [...] clear and compelling*”, and the approach offered “*a good idea for the important field of network inference.*”; the reviewer concludes that there are a “*number of important issues [to be] clarified*” before the reviewer “*can recommend publication.*”

Reviewer 2 praises the “*new method*” introduced, identifies that “*the proposed method outperforms*” alternative methods that in principle are suitable under identical conditions. The reviewer emphasizes that “*the results presented with the AUC are in the highest standards [the reviewer has] seen among the papers on network reconstructions. Thus, researches following this work should regard this work as a role model. Therefore, practically, this work is appealing and should be published.*” The reviewer specifically asks us to more clearly present some of the technical details, assumptions made and expand on the related literature.

Reviewer 3 underlines that we “*propose a model-free method for inferring interactions of system elements (or units) of the 2nd or even higher order.*” and that “*The problem is definitely important in the study of non-linear dynamic system.*” That reviewer suggests to also clearly explain differences to existing related approaches, to clarify some technical details, and specifically asks whether there is “*any specific real application that can be used to demonstrate the usefulness of the proposed method? for example, in inference of gene regulatory relationship...*”

In our replies below and our major revision of some parts of the manuscript, we addressed all of the reviewers’ comments. Some of the comments may have been based on a misunderstanding and we have improved clarity of the relevant manuscript text where appropriate.

In the revised manuscript, we now more clearly defined the assumptions and some more technical details, further clarified the presentation and introduced a new subsection demonstrating for two biological system settings that our approach can be applied, is successful as for the artificial model settings and indeed also outperforms existing approaches operating under the same conditions.

Additionally, we have added three new sections to the Supplementary Information where we first summarize how our approach scales with respect to network size, number of incoming connections and noise level and determine the complexity of our algorithm. Second, we explain in more detail the biological models systems employed in the main manuscript. And third, we provide a brief explanation of the codes for simulating and reconstructing networks from time series, which we are enclosing with this resubmission.

REPLIES TO THE COMMENTS OF REVIEWER 1

Reviewer 1 finds that the work is “*interesting*”, “*the analyses and Figures [...] clear and compelling*”, and the approach offered “*a good idea for the important field of network inference.*”; the reviewer concludes that there are a “*number of important issues [to be] clarified*” before the reviewer “*can recommend publication.*”. Specifically, the reviewer asks us to clarify where the main differences to our previous article lie and what the exact role of the basis functions in our current work is. We thank the reviewer for carefully reviewing the manuscript and for the useful comments. We address all comments and questions on a point-by-point basis in the following.

Comment: “*This paper explores sparse methods to identify networks of dynamical interactions. The authors use group sparse regression in a framework that identifies higher order interactions. They also demonstrate the method on a number of examples, investigating the learning rate and comparing with other leading methods. The paper is interesting, and the analyses and Figures are clear and compelling. Identifying network structure with sparse methods is a good idea for the important field of network inference. However, there are a number of important issues that need clarification before I can recommend publication.*”

Reply: We thank the reviewer for the positive assessment about our work and its potential impact on the network inference field. We hope that the replies provided below resolve all the referee’s concerns.

Comment: “*It is important to clarify the key differences between the present work and the authors’ recent Science Advances paper. They seem extremely close in theory... both identify networks using compressed sensing? This is important to clarify to assess if the present work is sufficiently novel.*”

Reply: We thank the referee for expressing his concern. Both contributions indeed address the problem of network reconstruction from collective dynamics. However, there are a number of major conceptual differences that are also of relevance for applications: The contribution in Science Advances exploits (i) time-averages of (ii) stationary responses of (iii) sparsely-connected networks to (iv) known driving signals using (v) compressed sensing. Our submission to Nature Communications exploits (i) the full trajectory of (ii) transient, stochastic or other sufficiently complex dynamics of (iii) networks that need not be sparse. Moreover, the approach presented in the current submission works for systems that (iv) need not be driven, a property that we consider a substantial advantage for applications because a range of experimental conditions often do not allow controlled (and thus known) driving of the system. Finally, the approach presented in the current submission solve the resulting regression problems using (v) an algorithm based on the Block Orthogonal Least Squares algorithm, a computational scheme that does not rely on the minimization of $\|\cdot\|_1$, in contrast to common compressed sensing frameworks. These five differences are summarized in the following table:

Feature	Sci. Adv. contribution [1]	Nat. Comm. submission
Collective dynamics (I)	Averages of trajectory points	Full individual trajectories
Collective dynamics (II)	Non-transient, stationary	Transient or stochastic
Requires sparsity	Yes	No
Requires known drivings	Yes	No
Optimization technique based on...	Compressed sensing	Block-Orthogonal Least Squares

Table I. Differences between our Science Advances contribution and our Nature Communications submission.

In summary, our current submission to Nature Communications can be seen as an orthogonal framework to the contribution published in Science Advances. The new submission thus presents a novel theoretical concept and opens up a wider range of potential applications, in particular because of the option to reconstruct connectivity without the requirement to externally driving the system in a controlled way.

The specific properties and requirements of the new approach are explicated in the revised version of the current submission.

Comment: “*Can you clarify: does this method identify the graph of connections but not the specific nonlinear terms in the dynamics? Can it identify the terms once the graph structure is identified?*”

Reply: We thank the reviewer for this relevant question. We designed the method to recover the physical network interactions in terms of its dependency structure (graph). Moreover, if the basis functions match the specific coupling functions in the original system, our approach would in addition identify the exact nonlinear terms, generalizing the findings in, e.g., [2] and [3]. Nevertheless, we expect that more often than not experimentally accessible networked

systems exhibit complicated coupling functions that can only be approximated by the linear combinations of basis functions or at least are not exactly known.

In the revised manuscript, we have further clarified that our approach is meant to reconstruct network interactions in terms of physical inter-unit dependencies.

Revised: We remark that here we do not intend to recover the actual functional form of interactions, but instead we aim at determining the existence of such interactions.

Comment: *"I believe that the code to generate and analyze data should be included along with the manuscript for the referees and for the readers. This will make the method much more transparent, and will promote reproducible research."*

Reply: We thank the reviewer for this note and agree that making the code available is valuable. Along with the revised manuscript, we are now providing a complete set of codes for simulating and reconstructing the connectivity of networks of several dynamical systems. To illustrate the full procedure from simulation to evaluating the reconstruction, we also provide example files. The examples are also presented in the Supplementary Information.

Comment: *"How does the method perform if the system is sampled without any transients. I can imagine many systems where the dynamics/interactions could be ambiguous on a limit cycle, but where in fact there are hidden variables and connections that only appear with transients. This might be clarified in the paper."*

Reply: We agree with the reviewer on that interactions could be ambiguous if the dynamics is restricted to special, lower-dimensional sets, for instance to a simple limit cycle. Indeed, Reviewer 2 also commented on a similar issue. Our approach relies on sampling the dynamics space. So, more extensive samplings of such space promote higher-quality reconstructions. If the system dynamics is restricted to a low-dimensional sets (invariant or not), indeed no information about some or all network interactions might be available, such that no method in principle could reveal them from recorded time series alone. If, on the other hand, stochastically driven, transient or otherwise sufficiently complex collective dynamics is observed, reconstruction becomes possible, because the dynamics space is sampled sufficiently. We thoroughly adapted the statements about the conditions for reconstruction in the revised manuscript. In particular, we have added the following text to the discussion of the revised manuscript:

Revised: In particular, collective dynamics that is transient, stochastically driven, or otherwise sufficiently complex helps revealing interactions whereas certain stable dynamics on low-dimensional subsets of state space only sample limited regions of the dynamics space and thus in principle do not provide full information about network interactions.

Comment: *"This work seems quite closely connected to related methods that identify dynamics with sparse methods, such as references [14,32]. You mention these methods on page 6, and only in passing to say that this method is fundamentally different because it doesn't require exact representation in a basis or need to admit a sparse representation. However, it really doesn't look very different, as it relies on the choice of a basis "h" and then uses sparsity to identify connections. I am not saying that it is the same, but to brush over the strong similarities seems like a mistake. In fact, this does not strengthen the impact of the present work, but isolates it. Also, Reference [14] is mischaracterized in the introduction (line 34): this method does not only determine unknown parameters, but identifies underlying connections and nonlinear dynamics."*

Reply: We thank the reviewer for explicating this issue. Indeed, there are similarities of our work to that reported in the cited references, yet these are much weaker than they may seem at first, because (a) the basis functions in the literature, did exactly fit the underlying model, a requirement not necessary for our approach, and moreover (b) we in fact do not use sparsity as a main precondition, but block-orthogonal least squares method. Let us explain in detail. As demonstrated in section **Proper basis functions and learning curves**, our method does not depend on the actual h chosen, but on the number of variables h takes as arguments, e.g. $h(x)$ for one, $h(x, y)$ arguments, and so on. So, for instance, a bivariate function $s(x, y)$ is only correctly represented in a bivariate function space spanned by functions $h_p(x, y)$, with $p \in \{1, 2, \dots, P\}$, yet not in the union of two univariate function spaces spanned by functions $h_p(x)$ and $h_p(y)$. We illustrate this idea in Figure 4, where we show that networks with complicated bivariate coupling functions (detailed in 'Networks and Hypernetworks of phase-coupled oscillators' in the Online Methods) are correctly reconstructed by four different families of bivariate basis functions $h_p(x, y)$, $p \in \{1, 2, \dots, P\}$. Concerning sparsity, we do not use the sparsity constraint as done in our Science Advances and other works exploiting the idea of compressed sensing. We only assume that zero entries, if any, appear in groups and propose an algorithm to solve such kind of systems by block-orthogonal least squares techniques, without resorting to compressed sensing techniques. Nevertheless, regarding reference [14], we have modified the text in the introduction of the main manuscript, and now it reads:

Revised: Current nonlinear dynamics approaches may recover direct interactions from the collective dynamics of a system if a mathematical model is provided in advance and only their unknown parameters, network links and nonlinear terms are to be determined [10-18].

Comment: "A recent paper by Mangan et al (*IEEE Transactions on Molecular, Biological... "Inferring biological networks by sparse identification..."*) uses related sparse methods to identify networks for biological models. These methods identify Michaelis Menten kinetics and yeast glycolysis networks (both methods that you identify/mention). This is an important reference to discuss/compare."

Reply: We thank the reviewer for pointing out this work, we were not aware of this contribution. We added the reference and the following text in the discussion:

Revised: Moreover, a work [Mangan et al., 2016] applying an extension of the method described in [Brunton et al., 2016] on models for gene regulation also suggests that such approaches scale supra-linearly with the dimensionality of the network, for both the number of candidate coupling functions and the time points necessary for successful reconstruction.

Comment: "There are statements in the discussion about how this method scales better to large networks. Without including code or a computational complexity analysis, it is unclear if the proposed method actually scales to larger systems better than alternative methods."

Reply: We thank the reviewer for this suggestion. We agree with the reviewer that the scaling of the method might not have been explained in sufficient detail in the discussion. In the manuscript, we refer to two classes of scaling that is essential for applications: (i) given that reconstructing the input connectivity of a unit is mathematically independent of the reconstruction for any other unit, we may reconstruct a full network by reconstructing the connectivity of each unit in parallel; (ii) there is a scaling with respect to the necessary number of data points for successful reconstructions as a function of system size and other factors. Figure 3a-b of the manuscript shows that the necessary number of data points for successful reconstruction increases linearly with the number of incoming connections and sub-linearly with the system size. In the revised manuscript, we modified the discussion and added the following text to the discussion section:

Revised: The theory presented above does neither require prior knowledge of parameters and coupling functions involved in the network dynamics nor does it require these functions to admit a sparse representations in any basis chosen; it is not limited to low-dimensional networked systems, and the number of necessary time points for successful reconstruction scales sublinearly with network size.

Regarding the algorithm, we explicate in the revised Supplementary Information that our method scales, in the worst case, quadratically with the number of coupling functions per unit. (compare our replies to the previous comment).

Comment: "How does this method scale to high dimensional systems? What is the highest dimension tested?"

Reply: As shown in Figure 3a, the method scales sublinearly with the system size (see also our replies above). In the results reported in this original submission, the largest size tested is $N = 1000$. Reconstructing the input connectivity of a unit in such a network of $N = 1000$ units takes about one minute on a single standard computer. We have added the following statement about absolute time performance in the revised manuscript:

Revised: Moreover, inferring the incoming connections of single units in large sparse networks ($N = 1000$, $n_i = 10$) in conventional hardware (Intel® Core™ i5-2430M) takes 65 ± 26 s per unit. Such results highlight the potential applicability of our approach in combination with parallel computing for revealing interactions in real-world networks, which are often large in size and sparsely connected.

Comment: "How would this method work if the dynamical interactions are more complex (sinusoidal, or other generalized functions)? What if the dynamics functions have no support in the basis you choose for "h"? Is the choice of basis explored, and multiple bases compared?"

Reply: We thank the reviewer for these questions about the models and bases. To improve readability, we have now included a short description of all models used in this (re)submission in the Online Methods section. All models employed have non-trivial coupling functions ranging from sinusoidal functions having one and two Fourier modes to Michaelis-Menten potentials.

The choice of basis is explored in the main text and illustrated in Figure 4. The results demonstrate that learning curves with respect to the number of inferred interactions are a visual help to assess the appropriateness of chosen basis functions – solely based on the recorded time series. In particular, we studied the behavior of learning curves for six different basis functions for a network of coupled phase-oscillators with two bivariate coupling functions (see

also 'Networks and Hypernetworks of phase-coupled oscillators' in the Online Methods). The Figure shows that all bivariate bases show an L-shaped learning curve with a plateau at the correct number of incoming interactions, while univariate bases show no plateau. These results indicate that learning curves reveal whether the dynamics has a support in the function space spanned by the selected basis functions.

Comment: "Typos:"

Reply: We thank the reviewer for carefully reading the manuscript and detailed suggestions. In what follows, we corrected the typos and modified the manuscript according to the reviewer comments.

Comment: "After equation 1, comma missing before " x_N " "

Reply: We have added the comma. The in-text math display now reads:

Revised: $\mathbf{x}(t) = [x_1(t), x_2(t), \dots, x_N(t)]$

Comment: "Same paragraph: Generally noise added to the state of a differential equation is referred to as a "disturbance" (as opposed to noise added to measurements that do not affect the dynamics of the system"

Reply: We thank the reviewer for this comment. The notions actually do depend on the research community considered. The large community of researchers on nonlinear dynamical systems as well as a large fraction of those performing computational modeling and analysis would use the wording "noise" as we do. We expect these two communities to constitute a significant part of the readership of our work. For this reason, we decided to keep this wording as is, being aware it is a compromise.

REPLIES TO THE COMMENTS OF REVIEWER 2

Reviewer 2 praises the "new method" introduced, identifies that "the proposed method outperforms" alternative methods that in principle are suitable under identical conditions. The reviewer emphasizes that "the results presented with the AUC are in the highest standards [the reviewer has] seen among the papers on network reconstructions. Thus, researches following this work should regard this work as a role model. Therefore, practically, this work is appealing and should be published." The reviewer specifically asks us to more clearly present some of the technical details, assumptions made and expand on the related literature. We thank the reviewer for carefully reviewing the manuscript and for the useful comments. We address all comments and questions on a point-by-point basis in the following.

Comment: "The authors propose a new method for inferring direct interactions from observed time series generated from coupled nonlinear systems. The key idea for the proposed method is to consider the superposition of general functions with an argument, two arguments, three arguments and forth to represent the underlying dynamics. Then, the authors fit the superposition by minimizing a projection error function. The performance for the proposed method outperforms those for correlations, partial correlations and transfer entropy. In addition, the results presented with the AUC are in the highest standards I have seen among the papers on network reconstructions. Thus, researches following this work should regard this work as a role model. Therefore, practically, this work is appealing and should be published."

Reply: We thank the reviewer for the positive assessment about our work and the statements that our submission should be a role model for future contributions in the field and that our submission should be published in Nature Communications.

Comment: "The authors do not declare their assumptions clearly, one of which is that a time series is generated by transient states or stochastic dynamics. Thus, the authors should emphasize such assumptions. If they also want to include a set of nonlinear deterministic systems on the attractors, they need to (a) cite more appropriate papers including Nawrath et al., *Phys. Rev. Lett.* (2010) and Zou et al., *Int. J. Bifurcat. Chaos* (2011), and (b) comment on the effect of synchronization (Palus and Vejmelka, *Phys. Rev. E*, 2007) as well as the effect of hidden common causes (Hirata and Aihara, *Phys. Rev. E*, 2010)."

Reply: We thank the reviewer for emphasizing this issue. A similar comment has also been made by reviewer 1 and we have revised the manuscript accordingly. Our approach relies on extensive samplings of the dynamics space. Thus, if the networked system resides in a synchronized state or some other lower-dimensional dynamics (e.g. fixed point or limit cycle), the time series by itself may not, in principle, provide full information about network interactions,

because it samples a too limited region (a low-dimensional subset) of the dynamics space. Under such conditions, added noise, long or multiple transients of otherwise sufficiently complex collective dynamics may help resolve the limited sampling problem. On the contrary, a system displaying an aperiodic dynamics (such as the case shown in Figure 1c) explores the dynamics space in a less-local manner, thus its dynamics provides more information for reconstructing the interactions. Hence, although transients do help in providing relevant information for reconstruction by easing the sampling of different locations in the dynamics space, it is not necessarily part of our core assumptions. In the revised manuscript, we have added the following text in the discussion:

Revised: In particular, collective dynamics that is transient, stochastically driven, or otherwise sufficiently complex helps revealing interactions whereas certain stable dynamics on low-dimensional subsets of state space only sample limited regions of the dynamics space and thus in principle do not provide full information about network interactions. Lower-dimensional dynamics may in particular be induced by symmetries or other invariants that can be reflected by conditions such as $z(\mathbf{x}) = 0$. For instance, systems evolving towards synchronized states, where the existence and directionality of interactions are impossible to extract from time series [4].

We further thank the reviewer for the references suggested, we included them in revised the manuscript where relevant.

Comment: "Their statement that one can represent uniquely the dynamics in the right hand side of Eq. (4) does not always seem to be valid. According to the paper by Judd and Nakamura, *Chaos* (2006), one cannot necessarily choose the correct model as the best model and some equivalent models exist. Even if we do not use any delays, simply imagine the case where there is some invariant equation represented by $z(x) = 0$. Then, after solving this invariant equation in terms of x if necessary, we can generate various equivalent equations. Thus the representations on the right hand side of Eq. (4) will become non-unique, and the "best" model might have an extra mechanism of noise reduction. Thus, the authors need to prove or assume that such an invariant equation does not exist for dynamics they analyze to ensure the uniqueness of the representations. This is another reason why the authors should emphasize their assumptions more."

Reply: We thank the reviewer for emphasizing this point. We agree with the reviewer on that one may not necessarily choose the *correct model* as the *best model* with respect to some optimization procedure and that equivalent models exist for a finite sampling of the dynamics space. First, if the dynamics is too low-dimensional or too simple, the collective dynamics in principle does not provide sufficient information to reveal a unique (and correct) model, independent of the approach. We have thus added a statement to clarify this issue (please see our revision text provided in response to the previous comment of the reviewer). Second, in principle, different choices of basis functions h combined with ARNI would produce different model estimates that are best in the above sense. Nevertheless, we here do not aim to provide a *best model* for the dynamics, but to provide a *correct set of input variables* (i.e. a set of input variables x_j for the possible $j \in \{1, \dots, N\}$ for a given unit i in Eq. (4)). As Figure 4 illustrates, if the number of arguments in the basis functions (e.g. $h(x)$ or $h(x, y)$) matches those of the interactions in the original system, our approach reveals the correct number and the correct set of input variables. In particular, the shape of learning curves shown in Figure 4 indicates whether the results obtained via the approach suggested in our work has the *correct set of input variables* or not. That we change the perspective and search for the set of units providing input to a given unit has been explicated already in the original version of the manuscript. To further clarify the issue, we have now added the following text to the introduction:

Revised: ... direct, physical interactions. To achieve this goal, we here change the perspective and ask which units j of the network provide direct physical interactions to a given unit i and appear in the right hand side of its differential equation, rather than asking for details of the interaction functions among those units.

Moreover, after discussing the basis functions and before evaluating performance, we now explicate that:

Revised: So, even if our scheme infers an optimal model from a given time series, it is not guaranteed that such a model would agree with an actual model generating the dynamics [5]. Indeed, the fact that we only ask for the units interacting with a given unit and not for details of the coupling functions enables robust performance across systems (compare Figures 3, 4, and 6).

We also agree with the reviewer that for some invariant equation $z(\mathbf{x}) = 0$ one may find a family of equivalent solutions. In our setup, this corresponds to the case where the networked systems are in a steady state ($\dot{x}_j=0$ for all $j \in \{1, \dots, N\}$). In this case, our method would fail to recover interactions among units because the recorded dynamics is only sampling a single point in the dynamics space. To avoid such complication, one may generate several transients to the steady state and reconstruct from such data sets as shown in Figure 2a-c.

Comment: "In Figure 3f, define more detail on how they define the AUC score when there are hidden elements. The score of more than 90% are too large if the half of the elements are hidden because then we cannot access about 75% of pairwise connections."

Reply: We agree with the reviewer and in the revised manuscript now state in more detail how we evaluate results for settings with hidden units. For such settings, we aim at correctly identifying existing and non-existing interactions only among the measured units, also if there are subsets of hidden units affecting the dynamics of the measured units. Thus, to compute AUC scores, we compare our predictions for the existence and absence of links among the measured units with those actually existing and absent among the measured units. The revised manuscript now states:

Revised: To compute AUC scores, we compare our predictions for the existence and absence of links among the measured units with those actually existing and not existing among those units, making no statement about indirect interactions mediated by hidden units.

Comment: "(iv) The references for partial correlations and transfer entropy are not shown appropriately. In addition, I had an impression that correlations, partial correlations, and transfer entropy the authors chose to compare are too old and not in the current state of the art. This point is obvious because they cited many recent methods but did not compare the proposed method with them quantitatively. Thus, if time allows, the authors might want to revise this point so that we can accelerate the researches in this field and enhance the discussions. But, this latter part is optional for the revision."

Reply: We thank the reviewer for pointing out that we did not properly cite partial correlations and transfer entropy. In the revised manuscript, we have added the appropriate citations. Concerning the benchmarks, we agree with the reviewer that there are newer tools for network reconstruction in general. However, we aimed at comparing our approach to model-independent methods. Many of the methods described in our introduction are model-dependent and others demand the dynamics to be measured with specific protocols (e.g. requiring external driving of the system) which are not compatible with the method described in our submission. Thus, we decided to use classical benchmarks which are also model-independent and compatible with the time series data sets and conditions employed in this work.

Comment: "Although the authors state in the main text and the title that their method is model-free, their performance depends on which basis functions they use (see Fig. S5). Thus, the title of "model-free inference" might be overstated and should be rephrased appropriately."

Reply: We thank the reviewer for this constructive criticism. Although the performance of our method depends on the family of basis functions employed, it does not depend on the specific choice of function itself. As shown in the section *Proper basis functions and learning curves*, our method depends on the number of variables used in the basis functions h , e.g. $h(x)$ and $h(x, y)$. This idea is demonstrated in Figure 4, where we correctly reconstruct networks with complicated bivariate coupling functions (see also 'Networks and Hypernetworks of phase-coupled oscillators' in Online Methods) with four different types of bivariate basis functions. Furthermore, we show that the shape of learning curves with respect to the number of inferred interactions provide a criterion for selecting an appropriate family (e.g. univariate, bivariate, and so on) of basis functions. (Please also see our replies to a previous comment above.) Thus, no specific basis functions and therefore no specific model has to be provided or assumed in advance to be able to reveal connections via the method proposed in the manuscript. The several changes named above, about the assumptions about interaction functions, basis functions and graph links (directed pairs of units reflecting the presence of a direct interaction), should clarify this point in the revised manuscript.

REPLIES TO THE COMMENTS OF REVIEWER 3

Reviewer 3 underlines that we "propose a model-free method for inferring interactions of system elements (or units) of the 2nd or even higher order." and that "The problem is definitely important in the study of non-linear dynamic system." That reviewer suggests to also clearly explain differences to existing related approaches, to clarify some technical details, and specifically asks whether there is "any specific real application that can be used to demonstrate the usefulness of the proposed method? for example, in inference of gene regulatory relationship...". We thank the reviewer for carefully reviewing the manuscript and for the useful comments. We address all comments and questions on a point-by-point basis in the following.

Comment: "In this paper, the authors propose a model-free method for inferring interactions of system elements (or units) of the 2nd or even higher order. The problem is definitely important in the study of non-linear dynamic system. Given a non-linear dynamic system $d(x_i)/d(t) = F_i(x(t)) + \eta(t)$, the authors determine the interaction relationships by decomposing $F_i(x(t))$ into a (finite) series at the given time points: $d(x_i)/d(t) = \sum_j \Lambda_{jj}^i g_j^i(x_j) + \sum_{j,s} \Lambda_{jj}^i \Lambda_{ss}^i g_{j,s}^i(x_j, x_s) + \dots$ where $g_j^i(\cdot)$, $g_{j,s}^i(\cdot)$ are a LINEAR COMBINATIONS of some basis functions (listed in table 1, for example). Additionally, the authors give a method for such a decomposition. "

Reply: We thank the reviewer for the detailed assessment of both our work and the relevance of the problem in the field. Our replies provided below as well as our revisions of the manuscript addressed all the issues raised and questions asked by the reviewer.

Comment: *"Studying the properties of a function through decomposition is not new, for which an exemplar is Fourier analysis or wavelet."*

Reply: We agree with the reviewer on that analyzing the properties of functions by decomposing them in simpler functions is not new. However, here rather than studying a function's properties, we provide an approach to determine the actual subset of input variables x_j for $j \in \{1, \dots, N\}$ on the right hand side of a system of ordinary differential equations representing the dynamics of a unit i of a network dynamical system. This differs from the classical perspective of Systems Identification [6] in the sense that we initially do not intend to provide a specific model to describe $F_i(\cdot)$, instead we only care the set of its about actual input variables. We use function decomposition as a tool to transform the input-identification problem into a multivariate regression problem with grouped variables, which we then solve with the algorithm proposed in the manuscript. As a similar question has also been asked by reviewer 2, we have complemented the manuscript text with an additional statement about this distinction already in the introduction. It reads:

Revised: ... direct, physical interactions. To achieve this goal, we here change the perspective and ask which units j of the network provide direct physical interactions to a given unit i and appear in the right hand side of its differential equation, rather than asking for details of the interaction functions among those units.

Comment: *"The authors do not answer the following questions regarding their method well: (a) Why are the basis functions in Table 1 used? (b) Are the decomposition with respect to the basis functions in Table 1 used in wavelet theory, or somewhere else?"*

Reply: We thank the reviewer for these questions about how we choose the basis functions. We used the basis functions shown in Table I to demonstrate that different families of basis functions are equally appropriate for revealing network interactions. In particular, we wanted to demonstrate in section **Proper basis functions and learning curves** that our approach does not depend on the specific choice of functions (Please also see our replies to a previous comments by reviewers 1 and 2).

Concerning our choices for testing our approach, our criteria were (i) simplicity and (ii) appropriateness for networked systems. Thus, we employed expansions in power series (bases a, b and e in Table I), Fourier series (c and f in Table I) and radial basis functions (d in Table I). In particular, bases a and c in Table I span function spaces with respect to the difference of the activity of units j and i . It is known that many networked systems exhibit diffusive interactions of the form $g_{ij}(x_i, x_j) = \tilde{g}_{ij}(x_j - x_i)$, for which bases a and c are appropriate options for basis functions. To our current understanding, such bases are not used in wavelet theory.

Comment: *"(c) is there any specific real application that can be used to demonstrate the usefulness of the proposed method? for example, in inference of gene regulatory relationship, which is mentioned several times in the manuscript."*

Reply: We thank the reviewer for raising this concern. To demonstrate the potential usefulness of our approach on biological systems, we originally reconstructed the connectivity of (i) models representing gene regulation (Figure 2a-c), and glycolytic oscillations in Yeast (Figure 2d-f in original manuscript). We slightly restructured the manuscript to achieve this goal. To avoid confusion:

- we replaced the glycolytic oscillations example by the reconstruction of networks of phase oscillators (Figure 2d-f in revised manuscript),
- and, included a new section titled **Robust inference of biological networks**, where we now demonstrate the efficiency of our approach on the glycolytic oscillator network in yeast and the circadian clock in *Drosophila*.

The new section reads:

Revised: Next we establish the potential of our framework to reconstruct interactions for biological system settings. Specifically, we demonstrate results on two networked model biological systems: glycolytic oscillator in yeast [7] and circadian clock in *Drosophila* [8]. The glycolytic oscillator, exhibiting one of the classical examples for cellular oscillations, accounts for the main reactions of glycolysis. Here we focus on a model for anaerobic glycolytic oscillations in yeast, containing the influx of glucose and outflux of pyruvate and/or acetaldehyde [7] (see Supplementary Information V for an extended description). The circadian clock underlies the biological response to the day-night cycle, and the oscillations it exhibits in *Drosophila* are driven by a negative feedback between two genes and the complex that is

formed by the proteins they code for. The model equations for the circadian clock are based on [8] (see Supplementary Information V).

Employing the above approach of combining a dynamics space representation, expanding in suitable families of basis functions and solving the resulting linear regression problem by an orthogonal least squares method, we reconstructed the interactions between the different components of the glycolytic oscillator (Figure 6a-b) and the circadian clock (Figure 6c-d) from transient dynamics towards their periodic orbits. As for the other systems' settings, the results confirm that larger number M of observations improve the predictions. Moreover, the reconstruction quality by this method again outperforms those resulting from correlations, partial correlations and transfer entropy.

Comment: *"Minor comments.*

The manuscript is prepared poorly. There are many grammar mistakes, ambiguous concept and awkward sentences. Some Figures are not understandable. Here is the list of some instances."

Reply: We thank the reviewer for his detailed and careful reading, and in what follows, we tackle all of the typos mentioned and clarified a few sentences and aspects of Figure captions.

Comment: *"Line 17, page 2. "amount" not "amounts"."*

Reply: We have made the change, the sentence now reads:

Revised: Accelerating technological progress creates a massive amount of available data about collective nonlinear dynamics in physical, biological and technological systems.

Comment: *"Line 39, page 2. units are not defined? what are units?"*

Reply: We have rephrased the sentence to indicate that we refer to network units (mathematically speaking these are the indices of the components of the state vector). The sentence now reads:

Revised: Common approaches construct functional links by detecting statistical dependencies (e.g. correlations, mutual information, Granger causality and extensions thereof) [19-27], and thus are prone to recover indirect relations among the units of a network, for instance, due to common external inputs or decorrelating effects induced by other units in the network [15, 23, 24].

Comment: *"Line 80, page 3. in (3), Λ_{jj}^i has unnecessary DOUBLE subscript. Λ_j^i is enough."*

Reply: We agree that the double subscript is redundant. Mathematically speaking, there is one matrix Λ^i for each unit i , the explicit dependency matrix we introduce to achieve a simple and systematic representation of pairwise and potentially higher-order (triplet etc.) interactions in terms of systems of ordinary differential equations. As a consequence, on the right hand side for each i , the matrix appears exactly once. As expansions contain individual terms and products of diagonal elements of these matrices, they are correctly indexed Λ_{jj}^i . Briefly speaking, the non-zero diagonal matrix elements Λ_{jj}^i pick out those variables j that interact with i by multiplying the state vector (x_1, \dots, x_N) .

Comment: *"Line 99, page 5. In formula (5), the term $\eta_i(t)$ is mysteriously missing from formula (4)"*

Reply: We thank the reviewer for drawing our attention to this typo. We have (i) replaced the term $\eta(t)$ by $\xi(t)$ to make notation consistent throughout, and (ii) included $\xi(t)$ and $\xi_i(t)$ where relevant.

Comment: *"Line 105, page 6. punctuation problem before "as" "*

Reply: By $h_{..}$ we mean to indicate that basis functions h have two subscripts separated by a comma (the left argument may be a multi-index).

Comment: *"Figure 2 is positioned correctly. Part (b) is not understandable. "a relaxation to a steady state" means what?"*

Reply: We thank the reviewer for his assessment about the Figure. We have replaced Figure 2b with a larger plot and added the following text to its description in the caption:

Revised: Example of transient dynamics towards steady states, where $\dot{x}_j = 0$ for all $j \in \{1, 2, \dots, N\}$.

The final positioning of all Figures will be handled when the final manuscript is layouted and typeset for production. We will take care of proper Figure positioning.

Comment: "Figure 5. The used network has 100 units. Why the Figure (b) and (c) were drawn for 40 units. how these 40 units were selected?"

Reply: We thank the reviewer for this questions. In Figure 5, we test the robustness of our approach against the existence of hidden units in the network. We mimic this condition by (i) simulating networks of $N = 100$, (ii) selecting just a fraction of 40 units of the network as the *recorded units*, and (iii) reconstructing the direct interactions only within the *recorded units*. The 40 recorded units are randomly selected. Figures 5b-c are give a qualitative picture of how the noise-free and noisy dynamics differ from each other. Also in response to a question by reviewer 2, we have added a sentence in the main text and modified the caption accordingly.

Comment: "Page 5, Supplementary document. Step 1 is "Select a generic model for interactions by choosing orders K, P_1, P_2, P_3 and so on, in (5)". P_1, P_2, P_3 are not found in (5) on page 5."

Reply: We thank the reviewer for drawing our attention to this typo. We meant to reference Equation (2) on page 5 [due to additions in the revised manuscript, Eq. (2) is now on page 6]. We have appropriately corrected the referencing in Step 1.

CHANGES IN THE SUPPLEMENTARY INFORMATION

New section on scalability of our approach:

Summary of Scaling Behavior

In what follows, we briefly summarize how the approach scales with respect to different intrinsic features of the networked systems, specifically, the system size, the number of incoming connections per unit, and the noise level in the time series. We also provide an explanation about a key aspect of the algorithmic complexity of our approach. As shown in the main manuscript, the number $M_{0.95}$ of required measurements to achieve reconstructions of AUC > 0.95 scales ...

- sublinearly, likely logarithmically $\sim \log(N)$ (see Figure 3a in main manuscript), with number N of dynamical variables in the network (that equals the number of units if they exhibit one-dimensional dynamics). Similar results have been reported in [9, 10].
- linearly, $\sim n_i$ (see Figure 3b in main manuscript), with the number n_i of incoming connections per unit.
- supralinearly (see Figure 3c in main manuscript) with the noise strength η .

Concerning the algorithmic complexity of our approach, if Q is the total number of proposed expansions in the right hand side of (5), it follows that the total number of projections for the worst-case condition $\mathcal{N} \setminus \hat{\mathcal{L}}_{i,(l-1)} = \emptyset$ is given by

$$\# \text{ Projections} = \underbrace{Q}_{1\text{st iter.}} + \underbrace{(Q-1)}_{2\text{nd iter.}} + \underbrace{(Q-2)}_{3\text{rd iter.}} + \dots + \underbrace{1}_{Q\text{-th iter.}} = \frac{Q(1+Q)}{2} \sim \mathcal{O}(Q^2). \quad (1)$$

Therefore, in the worst case where all Q expansions are needed to approximate the right hand side of (5) or the threshold θ is too small, our algorithm has a complexity of $\mathcal{O}(Q^2)$.

New section on biological model systems:

Biological model systems

To test performance on biological model systems, we simulated the glycolytic oscillator in yeast [7] and circadian clock in *Drosophila* [8] and inferred their interactions from their collective dynamics.

Glycolytic oscillator model

Glycolytic oscillations is one of the best-studied examples for oscillations at the cellular level. All its factors oscillate at different phases with a shared relatively short period. In yeast cells, these oscillations were also shown to synchronize

Parameter	Values
J_0	2.50 mM min ⁻¹
k_1	100.00 mM ⁻¹ min ⁻¹
k_2	6.00 mM ⁻¹ min ⁻¹
k_3	16.00 mM ⁻¹ min ⁻¹
k_4	100.00 mM ⁻¹ min ⁻¹
k_5	1.28 min ⁻¹
k_6	12.00 mM ⁻¹ min ⁻¹
k	1.80 min ⁻¹
κ	13.00 min ⁻¹
q	4.00
K_1	0.52 mM
ψ	0.10 mM
N	1.00 mM
A	4.00 mM

Table II. Parameters for glycolytic oscillator.

across cell population [11]. Here we use a single-cell model for anaerobic glycolytic oscillations in yeast, containing the influx of glucose and outflux of pyruvate and/or acetaldehyde, resulting in coupled ODE equations for seven species [7]

$$\dot{S}_1 = J_0 - \frac{k_1 S_1 S_6}{1 + (S_6/K_1)^q} \quad (2)$$

$$\dot{S}_2 = 2 \frac{k_1 S_1 S_6}{1 + (S_6/K_1)^q} - k_2 S_2 (N - S_5) - k_6 S_2 S_5 \quad (3)$$

$$\dot{S}_3 = k_2 S_2 (N - S_5) - k_3 S_3 (A - S_6) \quad (4)$$

$$\dot{S}_4 = k_3 S_3 (A - S_6) - k_4 S_4 S_5 - \kappa (S_4 - S_7) \quad (5)$$

$$\dot{S}_5 = k_2 S_2 (N - S_5) - k_4 S_4 S_5 - k_6 S_2 S_5 \quad (6)$$

$$\dot{S}_6 = -2 \frac{k_1 S_1 S_6}{1 + (S_6/K_1)^q} + 2k_3 S_3 (A - S_6) - k_5 S_6 \quad (7)$$

$$\dot{S}_7 = \psi \kappa (S_4 - S_7) - k S_7 \quad (8)$$

where S_1 represents glucose, S_2 represents glyceraldehydes-3-phosphate and dihydroxyacetone phosphate pool, S_2 represents 1,3-bisphosphoglycerate, S_2 represents cytosolic pyruvate and acetaldehyde pool, S_2 represents NADH, S_2 represents ATP, and S_7 represents extracellular pyruvate and acetaldehyde pool. Parameters were set according to [12] and are shown in II. Initial conditions were randomly selected from the uniform distribution $S_i(0) \in [1.2, 2.0]$ and employed $\Delta T = 0.01$ min.

Circadian Clock

Circadian clocks underly the response to the day-night cycle. Specifically, we simulated a model of the circadian clock in *Drosophila* [8], where oscillations are driven by a negative feedback between *per* and *tim* genes, which code for PER and TIM proteins, and the PER-TIM complex. The rate equations for the 10-species circadian clock are:

$$\dot{M}_P = v_{sP} \frac{K_{IP}^n}{K_{IP}^n + C_N^n} - v_{mP} \frac{M_P}{K_{mP} + M_P} - k_d M_P \quad (9a)$$

$$\dot{P}_0 = k_{sP} M_P - V_{1p} \frac{P_0}{K_{1P} + P_0} + V_{2p} \frac{P_1}{K_{2P} + P_1} - k_d P_0 \quad (9b)$$

$$\dot{P}_1 = V_{1p} \frac{P_0}{K_{1P} + P_0} - V_{2p} \frac{P_1}{K_{2P} + P_1} - V_{3p} \frac{P_1}{K_{3P} + P_1} + V_{4p} \frac{P_2}{K_{4P} + P_2} - k_d P_1 \quad (9c)$$

$$\dot{P}_2 = V_{3p} \frac{P_1}{K_{3P} + P_1} - V_{4p} \frac{P_2}{K_{4P} + P_2} - k_3 P_2 T_2 + k_4 C - v_{dP} \frac{P_2}{K_{dP} + P_2} - k_d P_2 \quad (9d)$$

$$\dot{M}_T = v_{sT} \frac{K_{IT}^n}{K_{IT}^n + C_N^n} - v_{mT} \frac{M_T}{K_{mT} + M_T} - k_d M_T \quad (9e)$$

$$\dot{T}_0 = k_{sT} M_T - V_{1T} \frac{T_0}{K_{1T} + T_0} + V_{2T} \frac{T_1}{K_{2T} + T_1} - k_d T_0 \quad (9f)$$

$$\dot{T}_1 = V_{1T} \frac{T_0}{K_{1T} + T_0} - V_{2T} \frac{T_1}{K_{2T} + T_1} - V_{3T} \frac{T_1}{K_{3T} + T_1} + V_{4T} \frac{T_2}{K_{4T} + T_2} - k_d T_1 \quad (9g)$$

$$\dot{T}_2 = V_{3T} \frac{T_1}{K_{3T} + T_1} - V_{4T} \frac{T_2}{K_{4T} + T_2} - k_3 P_2 T_2 + k_4 C - v_{dT} \frac{T_2}{K_{dT} + T_2} - k_d T_2 \quad (9h)$$

$$\dot{C} = k_3 P_2 T_2 - k_4 C - k_1 C - k_2 C_N - k_{dC} C \quad (9i)$$

$$\dot{C}_N = k_1 C - k_2 C_N - k_{dN} C_N \quad (9j)$$

where M_T and M_P are *tim* and *per* mRNAs, respectively. T_0 , T_1 and T_2 are forms of TIM protein, P_0 , P_1 and P_2 are forms of PER protein, and C and C_N are forms of PER-TIM complex.

The total levels of PER and TIM proteins, P_t and T_t , respectively, are given by:

$$P_t = P_0 + P_1 + P_2 + C + C_N \quad (10a)$$

$$T_t = T_0 + T_1 + T_2 + C + C_N \quad (10b)$$

The parameter values used for the simulations are based on [8].

New section on codes for simulation and reconstruction:

Codes for simulating and reconstructing networks

In addition to this Supplementary Information, we also provide a set of codes (including examples) for simulating time series and reconstructing the connectivity of different models of network dynamical systems. The codes we provide include:

```

ARNI_NatComm
├── Data
│   ├── connectivity.dat
│   ├── data.dat
│   ├── frequencies.dat
│   └── ts_param.dat
├── Functions
│   ├── basis_expansion.m
│   ├── reconstruct.m
│   ├── simulate.m
│   └── topology.m
├── Html_Output
├── Models
│   ├── kuramoto1.m
│   ├── kuramoto2.m
│   └── michaelis_menten.m

```

```

├── roessler.m
├── example1.m
├── example2.m
├── example3.m
└── example4.m

```

In what follows, we give a brief description of each file.

Data

The folder Data contains all the necessary information for reconstructing and evaluating the quality of reconstruction of networks from time series. Specifically, CONNECTIVITY.DAT contains the connectivity matrix of the network model simulated. The file DATA.DAT contains all the simulated time series for a specific simulated model. In case of simulating phase-coupled oscillators, FREQUENCIES.DAT contains the frequency of each oscillator. Finally, TS_PARAM.DAT indicates how many time series and for how long such were simulated.

Functions

basis_expansion.m

basis_expansion.m generates a multidimensional array of basis expansions evaluated on all points of a multivariate time series.

Output

Multidimensional array containing the evaluation of all basis functions for all time points and all possible incoming connections.

reconstruct.m

reconstruct.m returns a ranked list of the inferred incoming connections.

Output

LIST: Sequence of inferred interactions in the order such were detected.

COST: Fitting cost for all inferred interactions in the order such were detected.

FPR: False positives rate for the reconstruction.

TPR: True positives rate for the reconstruction.

AUC: Quality of reconstruction measured in AUC scores.

simulate.m

simulate.m generates time series of networks of dynamical systems for several different initial conditions.

Output

'DATA/DATA.DAT': File containing all simulated time series in a concatenated form.

'DATA/TS_PARAM.DAT': File containing time series parameters, i.e. number and length of time series.

topology.m

topology.m generates connectivity matrices for network simulation.

Output

'DATA/CONNECTIVITY.DAT': File containing a weighted adjacency matrix.

We refer the reader to the headers of each function (and the examples) for more details about the proper usage of these functions.

Html_Output

This folder contains the examples and their outputs in HTML format for simple access.

Models

This folder contains the files for simulating random ensembles of gene regulatory circuits (16), phase-coupled oscillators (17) with one and two fourier modes and networks of Rössler oscillators (19).

Examples

EXAMPLE1.M generates different time series of networks of dynamical systems starting from different initial conditions and reconstructs the connectivity for a selected unit. Increasing the number of different time series leads to better results.

Output Figure showing: (1) evolution of fitting costs versus the number of inferred interactions with actual inferred interactions; and, (2) Receiver-Operating-Characteristic Curve.

EXAMPLE2.M generates different time series for two different dynamical systems under different types of coupling functions, $h(x_j)$ and $h(x_i, x_j)$, starting from different initial conditions and reconstructs the connectivity for a selected unit. Increasing the number of different time series leads to better results. Bivariate coupling functions are only correctly represented by bivariate coupling functions. Analogously, univariate functions are only correctly represented by univariate basis functions.

Output Figures showing the evolution of fitting costs versus the number of inferred interactions using different bases for models kuramoto2 and michaelis_menten.

EXAMPLE3.M generates different time series for kuramoto2 systems and reconstructs them under radial basis functions of different orders. Greater orders (number of employed basis functions) lead to better results.

Output Figures showing the evolution of fitting costs versus the number of inferred interactions using different number of bases.

EXAMPLE4.M compares the quality of reconstruction between short and long time series with poor temporal resolution on networks of coupled chaotic roessler systems. Several short time series are preferable over long time series for this type of systems.

Output Figures showing the evolution of fitting costs versus the number of inferred interactions using different number of bases on kuramoto2 models.

-
- [1] Nitzan, M., Casadiego, J. & Timme, M. Revealing physical interaction networks from statistics of collective dynamics. *Sci. Adv.* **3** (2017).
 - [2] Shandilya, S. G. & Timme, M. Inferring network topology from complex dynamics. *New J. Phys.* **13**, 013004 (2011).
 - [3] Brunton, S. L., Proctor, J. L. & Kutz, J. N. Discovering governing equations from data by sparse identification of nonlinear dynamical systems. *Proc. Natl. Acad. Sci. U. S. A.* **113**, 3932–7 (2016).
 - [4] Paluš, M. & Vejmelka, M. Directionality of coupling from bivariate time series: How to avoid false causalities and missed connections. *Phys. Rev. E* **75**, 056211 (2007).
 - [5] Judd, K. & Nakamura, T. Degeneracy of time series models: The best model is not always the correct model. *Chaos* **16**, 033105 (2006).
 - [6] Ljung, L. *System Identification*. Advanced Textbooks in Control and Signal Processing (Springer London, London, 1999). arXiv:1011.1669v3.
 - [7] Wolf, J. & Heinrich, R. Effect of cellular interaction on glycolytic oscillations in yeast: a theoretical investigation. *Biochem. J.* **345**, 321–34 (2000).
 - [8] Leloup, J.-C. & Goldbeter, A. Chaos and Bifurcations in a Model for Circadian Oscillations of the PER and TIM Proteins in *Drosophila*. *J. Theor. Biol.* **198**, 445–459 (1999).
 - [9] Timme, M. Revealing Network Connectivity from Response Dynamics. *Phys. Rev. Lett.* **98**, 224101 (2007).
 - [10] Timme, M. & Casadiego, J. Revealing networks from dynamics: an introduction. *J. Phys. A Math. Theor.* **47**, 343001 (2014). 1408.2963.
 - [11] Richard, P. The rhythm of yeast. *FEMS Microbiol. Rev.* **27**, 547–557 (2003).
 - [12] Daniels, B. C. & Nemenman, I. Automated adaptive inference of coarse-grained dynamical models in systems biology. *Nat. Commun.* **6**, 38 (2014).

REVIEWERS' COMMENTS:

Reviewer #1 (Remarks to the Author):

I would like to thank the authors for a careful revision. I believe that they have addressed all of my concerns, and I can recommend the paper for publication.

Reviewer #2 (Remarks to the Author):

The authors have taken into account all of my concerns and revised the manuscript in a satisfactory manner. Thus, I recommend its publication.

Reviewer #3 (Remarks to the Author):

The authors revised the manuscript by taking the reviewers' suggestions into account. The issues raised by reviewers were addressed relatively well. However, some problems still remain.

The reviewer's commented that "this work seems quite closely connected to related methods that identify dynamics with sparse methods, such as references [14,32]". Indeed, the first part of decomposing the right-hand side of the dynamic system under study is identical to that in reference 32 (35 in the revised version). The only different is that the present work does not assume sparsity for dynamic systems under study. In other words, the authors apply an existing method in compressed sensing to general dynamic systems. As such, the contributions of this work only include:

(1) linking interaction between functional units of a dynamic system with the decomposition of the right-hand side of the dynamic system into selected basis functions.

and (2) the Algorithm for Revealing Network Interactions

(ARNI) for the decomposition. The first point is reasonable, but has no theoretic support. For the second point, the performance is only analyzed through simulation (summarized in Figure 4).

The contributions are valuable, but probably not enough for winning a spot in Nature Communications or PRL.

Figure 4 shows the performances of the algorithm for different sets of basis functions listed in Table I. What about the differences of the interactions inferred in difference cases?

It is good to see that the authors validated their method on two biological systems, despite they are not gene regulatory networks. The authors examined different methods using AUC scores (Figure 6). It would be much more clear if authors could discuss which true links in Figures 6.a and 6.c were correctly inferred and which were missing.

Lastly, there are still some ambiguous and awkward sentences. English editing is needed.

Minor comments

1. Do $M = S \times m$ in Figure 2 and M in Figure 3 have different meanings?

GENERAL COMMENTS BY THE REVIEWERS

We are grateful to the reviewers for again carefully reviewing our work. All reviewers positively judge our revisions. Reviewer 1 and Reviewer 2 both recommend publication as is. Reviewer 3 raises some remaining issues.

Specifically, Reviewer 1 explains that “[the authors] have addressed all of my concerns”, and recommends our submission for publication.

Reviewer 2 explains that “the authors have taken into account all of my concerns and revised the manuscript in a satisfactory manner”, and also recommends our submission for publication.

Reviewer 3 finds that “issues raised by reviewers were addressed relatively well” in our revised manuscript, and appreciates that we “validated [our] method on two biological systems...”. The reviewer has some remaining points and one remaining question.

In our replies below, we addressed the comments by Reviewer 3 after the second round of review. We believe the comments may have been based on a misunderstanding, and thereby we further explicate how our main contributions differ from previous studies. In particular, the biological systems illustrating our findings are indeed examples of gene regulatory networks, as asked for by reviewer #3.

REPLIES TO THE COMMENTS OF REVIEWER 3

Reviewer 3 finds that “issues raised by reviewers were addressed relatively well” and appreciates that we “validated [our] method on two biological systems...” in our revised submission; the reviewer suggests some amendments and clarifications. We thank the reviewer for their careful review of our revised manuscript, and in what follows, we address the issues raised by the reviewer on a point-by-point basis.

Comment: “[Reviewer 1] commented that “this work seems quite closely connected to related methods that identify dynamics with sparse methods, such as references [14,32]. Indeed, the first part of decomposing the right-hand side of the dynamic system under study is identical to that in reference 32 (35 in the revised version). The only different is that the present work does not assume sparsity for dynamic systems under study. In other words, the authors apply an existing method in compressed sensing to general dynamic systems.””

Reply: We thank the reviewer for expressing their concern. The reply provided below shall resolve the reviewer’s concern about the conceptual advances of our approach over the state-of-art.

At first glance the decomposition proposed in our submission may resemble that of [1], yet there is one key conceptual factor distinguishing our decomposition: in [1], the basis functions exactly match the functions present in the model to be reconstructed. So, if the basis functions do not match those in the model, the inference of the correct set of interactions is not guaranteed.

In contrast, in our approach, there is no need for basis functions to exactly match the interactions in the underlying system. In particular, we expect that single terms in the decomposition do not exactly fit the interactions, instead we expect that collections of linear combinations of basis functions approximate each of the interactions in the underlying model. Such conceptual idea makes our approach more general and robust by not relying on a specific set of basis functions. This is demonstrated in section **Proper basis functions and learning curves**, where we show that our approach does not depend on the actual h chosen, but on the number of variables h takes as arguments, e.g. $h(x)$ for one, $h(x, y)$ for two arguments, and so on. Specifically, we explicate this idea in Figure 4, where we reconstruct

- -

- - - - -

networks with complicated bivariate coupling functions (detailed in 'Networks and Hypernetworks of phase-coupled oscillators' in the Methods section) with four different families of bivariate basis functions $h_p(x, y)$, $p \in \{1, 2, \dots, P\}$. Thus, our contribution goes beyond *the application of an existing method* by (i) introducing a decomposition that is independent of the basis functions employed, and (ii) developing a suitable inference algorithm based on the systematic decomposition using explicit dependency matrices, and (iii) not requiring any sparsity but exploiting block structure only. So, we consider that our contribution presents a novel theoretical concept in the field of model-independent network inference from dynamics.

Comment: *"The contributions of this work only include: (1) linking interaction between functional units of a dynamic system with the decomposition of the right-hand side of the dynamic system into selected basis functions. and (2) the Algorithm for Revealing Network Interactions (ARNI) for the decomposition. The first point is reasonable, but has no theoretic support. For the second point, the performance is only analyzed through simulation (summarized in Figure 4)."*

Reply: We thank the reviewer for emphasizing these points. Concerning (1), as explained in the previous point, our contribution goes beyond proposing a decomposition that links the function of different dynamical units by introducing a decomposition that does not depend on a specific set of basis functions. Concerning (2), we chose to analyze the performance on *in-silico* systems to assess the predictive power of our approach in controlled settings. In this way, we could isolate and test the effect of different factors (e.g. network size, mean indegree, noise, hypernetwork interactions) on the performance of our approach.

We agree with the reviewer that there is no mathematical proof or theoretical derivation pinning down that this approach would always work, or the exact conditions when it may break down. By (i) newly introducing the systematic decomposition (using explicit dependency matrices), (ii) providing a new algorithm to handle the resulting inference problem, also (iii) without sparsity assumptions, and (iv) computationally testing the performance across abstract and specific model systems, we thereby achieve a model-independent framework in the first place. We believe it is reasonable that there are further open theoretical questions to be addressed in the future. Indeed, our core assumption is that the dynamical units are connected with each other. So, if such units are indeed connected, our approach may reveal their connectivity patterns purely from their dynamics without knowing a model beforehand.

Comment: *"Figure 4 shows the performances of the algorithm for different sets of basis functions listed in Table I. What about the differences of the interactions inferred in difference cases?"*

Reply: In Figure 4, we analyzed the effect of using univariate and bivariate basis to approximate complicated (and unknown) bivariate coupling functions. We observed that bivariate basis functions exhibit L-shaped learning curves with a clear plateau starting at the correct number of incoming interactions, point at which the algorithm has recovered all correct interactions. Thus, in cases (a-d), all the recovered links are identical. In particular, such learning curves are handy when there is no ground truth available to compare the inferences with.

Comment: *"It is good to see that the authors validated their method on two biological systems, despite they are not gene regulatory networks."*

Reply: We thank the reviewer for appreciating the new section about inferring the connectivity of biological models. Concerning the statement about gene regulatory networks, we believe there is a misunderstanding. We demonstrated the efficiency of our approach on biological network models that represent gene regulatory networks (please see details under 'Gene regulatory circuits' in the Methods section) in two separate sections in the manuscript. The first in Figure 2, where we evaluate the performance of our approach on such models with respect to the total number of time points. The second in Figure 5, where we study the more challenging case in which dynamical noise and hidden units (i.e. units that we assume are not available to measurement devices) act simultaneously on the observable units. In both cases, we show how the quality of reconstruction improves with increasing number of time points, and compare our results to other model-independent benchmarks.

Comment: *"The authors examined different methods using AUC scores (Figure 6). It would be much more clear if authors could discuss which true links in Figures 6.a and 6.c were correctly inferred and which were missing."*

Reply: We agree with the reviewer that determining the advantages and limitations of each benchmark in revealing specific links in the model systems may be of interest. Given that the result of directly comparing correct and incorrect identifications in an absolute manner would depend on settings of the system in combination with parameters of the algorithm (this is why we considered AUC curves in the manuscript), we consider that such extensive additional analysis is beyond the scope of the present manuscript that introduces the new concept in the first place.

Comment: *"Minor comments: 1. Do $M = S \times m$ in Figure 2 and M in Figure 3 have different meanings?"*

Reply: We thank the reviewer for this note. The quantity M always refers to the number of time points across the manuscript. In particular, for instance, $M_{0.95}$ represents the number of time points necessary to reach a reconstruction quality equal or larger than 0.95. If there is more than one piece of trajectory that is used for inference, we write $M = S \times m$, factorizing into the number of pieces and the length of a piece. To be able to compare inference given many short with that given one longer piece of trajectory, we label the total number of time points by M throughout the manuscript.

-
- [1] Brunton, S. L., Proctor, J. L. & Kutz, J. N. Discovering governing equations from data by sparse identification of nonlinear dynamical systems. *Proc. Natl. Acad. Sci. U. S. A.* **113**, 3932–7 (2016).